# Tokenized Bandit for LLM Decoding and Alignment

**Suho Shin** [1]  **Chenghao Yang** [2]  **Haifeng Xu** [2]  **MohammadTaghi Hajiaghayi** [1]

## Abstract

We introduce the tokenized linear bandit (TLB) and multi-armed bandit (TMAB), variants of linear and stochastic multi-armed bandit problems inspired by LLM decoding and alignment. In these problems, at each round $t \in [T]$, a user submits a query (context), and the decision maker (DM) sequentially selects a token irrevocably from a token set. Once the sequence is complete, the DM observes a random utility from the user, whose expectation is presented by a sequence function mapping the chosen token sequence to a nonnegative real value that depends on the query.

In both problems, we first show that learning is impossible without any structure on the sequence function. We introduce a natural assumption, diminishing distance with more commons (DDMC), and propose algorithms with regret $\tilde{O}(L\sqrt{T})$ and $\tilde{O}(L\sqrt{T^{2/3}})$ for TLB and TMAB, respectively. As a side product, we obtain an (almost) optimality of the greedy decoding for LLM decoding algorithm under DDMC, which justifies the unresaonable effectiveness of greedy decoding in several tasks. This also has an immediate application to decoding-time LLM alignment, when the misaligned utility can be represented as the frozen LLM's utility and a linearly realizable latent function. We finally validate our algorithm's performance empirically as well as verify our assumptions using synthetic and real-world datasets.

## 1. Introduction

Large Language Models (LLMs), in the past few years, received significant attention from extensive areas with its ability to serve numerous tasks including question answering, image generation, code completion and reasoning (Brown et al., 2020; Thoppilan et al., 2022; Fried et al., 2022; Zheng et al., 2024). Recent success of commercial LLMs such as ChatGPT, Gemini, and Claude (Team et al., 2023; Achiam et al., 2023) gives evidence on the feasibility of LLMs as personal assistants, supporting individuals' daily life in decision making, problem solving, and planning.

To serve as a genuine personalized assistant, however, it is crucial to align the LLM's outcome with the designated human preference, which remains as a core challenge in LLMs (Mishra et al., 2021; Sanh et al., 2021; Shen et al., 2023). Recent works suggest that aligning an LLM's output with human preference can be partially addressed by fine-tuning the model with reinforcement learning from human feedback (RLHF) (Bai et al., 2022; Ouyang et al., 2022; Rafailov et al., 2024) (see Wang et al. (2024) for comprehensive exposition). This line of research, however, typically requires significant computational resources due to fine-tuning, extensive human labeling, and frequent model updates, rendering it less scalable in practical and rapidly changing environments or as a genuine personalized assistant tailored to each individual's preference.

Consequently, a more recent thread of studies proposes *decoding-time alignment* methods, aiming to align the LLM's output on the fly during inference time without fine-tuning the LLM (Josifoski et al., 2022; Huang et al., 2024; Mudgal et al., 2023). While these methods hold promise in terms of scalability, their theoretical underpinnings remain largely uncharted, despite some recent evidences (Chakraborty et al., 2024; Li et al., 2024; Shi et al., 2024b), and how it could sample-efficiently learn the preference from user feedback is yet questionable. Moreover, even the decoding algorithm itself lacks foundational understanding from theoretical perspective despite being central to the LLM's quality, even the simplest ones such as greedy decoding or beam search, beyond a few recent works (Finlayson et al., 2023; Chen et al., 2024).

In this work, we propose a fresh perspective in the decoding-time alignment as well as decoding algorithm for LLMs by framing the LLM alignment and decoding problem as variants of linear/multi-armed bandits framework whose utility is represented via a sequence function and when a decision maker faces an online problem to select the sequence irrevocably in a token-by-token manner. Overall, our contributions are three-folds:

[1]University of Maryland [2]University of Chicago. Correspondence to: Suho Shin <suhoshin@umd.edu>.

*Proceedings of the 42nd International Conference on Machine Learning*, Vancouver, Canada. PMLR 267, 2025. Copyright 2025 by the author(s).

- We present tokenized variants of the linear and multi-armed bandit problems, which turns out to be closely connected to bandit tree search (BTS) problem (Coquelin & Munos, 2007). We first show that the general problem setup fundamentally exhibits exponential dependency in the maximal length of token sequences for any algorithm.

- To address it, we provide a fairly reasonable assumption, dubbed *diminishing distance with more commons* (DDMC), and provide sublinear regret algorithms for both problems. This complements the submodularity, widely adopted in sequence function maximization.

- We explain how our algorithms can be applied in the decoding-time LLM alignment. We discuss further interesting implications in the static counterpart of our problem, which might be of independent interest, demonstrating the provably (near) optimality of the greedy decoding under DDMC structure, justifying empirical evidences (Song et al., 2024) on the effectiveness of greedy decoding in certain tasks. We finally provide empirical experiments that justify our assumptions and performance of our algorithms.

In what follows, we briefly introduce our contributions and implications therein. Section 2 formally introduces our general problem setup. We provide our results for tokenized linear bandit problem in Section 3. Section 4 presents our results for the tokenized multi-armed bandit problem. Implications and applications to decoding-time alignment, decoding are presented in Section 5, and experimental results are provided in Section 6. Related works, and all the proofs are deferred to the appendix due to page limits. Appendix B presents our results with weakened version of DDMC, combined with lookahead decoding, and Appendix C discusses how our problems are connected to the seminal bandit tree search (BTS) problem.

### 1.1. Overview of results, assumptions, and implications

We start with the most general problem setup that will subsume *tokenized linear bandit* (TLB) and *tokenized multi-armed bandit problem* (TMAB).[1] Each round $t \in [T]$, a user arrives and provides a *query* $x_t$ as a context, and a decision maker (DM) selects token sequence $\mathbf{y}_t \in \mathcal{V}^*$[2] one at a time from a fixed vocabulary $\mathcal{V}$ with $|\mathcal{V}| = n$. Once the token sequence is finalized, the DM receives a *random utility* $r_t(\mathbf{y}_t)$ from the user, capturing how well the produced text aligns with the user's preference. The random utility consists of

two parts: (i) sequence function $u_t(x_t, \mathbf{y}_t)$ that maps a pair of context and token sequence to nonnegative real value, and (ii) random noise $\eta_t$ analogous to the standard bandit problems. The benchmark is to obtain sublinear regret against the optimal sequence in hindsight per each round, *i.e.,* $\text{OPT}_t = \mathbb{E}\left[\max_{\mathbf{y} \in \mathcal{V}^*} r_t(\mathbf{y})\right]$ and $\text{OPT} = \sum_{t=1}^{T} \text{OPT}_t$. We assume that the output token sequence is restricted to be within length of $L$, and seeks for algorithms with regret linear on $L$ and sublinear on $T$.

In TLB, our utility function is linearly parameterized by $u_t(x_t, \mathbf{y}) = \langle \theta, e(x_t, \mathbf{y}) \rangle$ given an embedding function $e : \mathcal{V}^* \times \mathcal{V}^* \to \mathbb{R}^d$ that maps a query and a token sequence to a feature vector, which can be viewed as an embedding vector that a typical LLM returns. In TMAB, utility function could be arbitrary, but the context remains the same $x_t = x$ to pose the problem learnable. For both problems, DDMC assumption is defined as follows: for any two token sequences $\mathbf{y}, \mathbf{z} \in \mathcal{V}^*$ with the same length $|\mathbf{y}| = |\mathbf{z}|$ and a token $\tau \in \mathcal{V}$, it follows that

$$|u_t(x_t, \mathbf{y} : \tau) - u_t(x_t, \mathbf{z} : \tau)| \le |u_t(x_t, \mathbf{y}) - u_t(x_t, \mathbf{z})|,$$

where $\mathbf{y} : \tau$ denotes the sequential concatenation of $\mathbf{y}$ and $\tau$ and similarly for $\mathbf{z}$. In words, user realizes small utility gap if two outputs have more common tokens (words) in suffix.

For TLB problem, we propose an excessive optimism under the face of uncertainty (EOFUL) algorithm with regret $O(L\sqrt{T \log T})$,[3] by combining greedy decoding with standard LinUCB algorithm (Abbasi-Yadkori et al., 2011). The analysis follows from several interesting ideas: (i) an introduction of level-$k$ regret that internally occurs from $k$-th chosen token during the algorithm, (ii) fictitious extension of our algorithm's sequence to recursively propagate the regret into upper level, followed by (iii) regret decomposition and sum of squares regret analogous to the standard linear bandit analysis. For TMAB problem, we no longer have a global estimate $\theta$ that will help us estimating the performance of each intermediate token sequence. Instead, we present GreedyETC that step-by-step learns a desirable path to efficient token sequences by combining exploration-then-commit style of algorithm with greedy decoding, inducing the regret $O(nLT^{2/3}(\log T)^{1/3})$. Analysis requires a careful union bound over the concentration of desirable token sequences, since naively taking union bounds over every token sequence yields regret of $\Omega(n^L)$.[4] We remark that DDMC assumption plays a crucial role, enabling certain tractable

---

[1]We note that TMAB does not directly reduce to TLB as a special case unlike the standard bandit problem, due to an exponential blow up in the dimension of the latent parameter. Thus, it requires independent algorithms and analysis.

[2]$A^*$ denotes a set of finite sequence over $A$.

[3]The analysis requires an additional technical assumption, detailed in Section 3.2. Linear realizability and this assumption is completely removed in our results for TMAB, at the cost of the same context across the rounds.

[4]TMAB problem, in fact, has a close connection to the well known *bandit tree search* (BTS), *i.e.,* a decision version of BTS can be reduced to TMAB and vice versa. We discuss the detail in Appendix C.

structure in the utility function, albeit its naturallity and simplicity.[5] DDMC assumption is relaxed in Appendix B, and we also provide efficient algorithms for both settings with the weakened assumption by replacing the greedy decoding with lookahead decoding, another well-known decoding algorithm (Lu et al., 2021; Fu et al., 2024).

Finally, we discuss several implications and applicability of our foundations in LLM decoding and alignment. For LLM alignment problem, we have a LLM equipped with a (known) utility function $v : \mathcal{V}^* \times \mathcal{V}^* \to \mathbb{R}_{\geq 0}$ and a misaligned utility represented by linear structure $\langle \theta, e(x_t, \mathbf{y}) \rangle$ with latent $\theta$, where the eventual utility is a unknown convex combination over them. The objective is to efficiently learn $\theta$ and the convex combination parameter in a decoding time through user feedback while leaving the original LLM frozen to ensure that no further training occurs for the LLM. We show that this problem reduces to the TLB problem, enabling efficient learning via EOFUL under DDMC. We next consider a static version of our problem, where a query access to the sequence function $u$ is available and we need to maximize $u$ with polynomial queries. Inspired by our analysis for EOFUL and GreedyUCB, we prove that the greedy algorithm achieves the exact optimum with $nLT$ queries. This immediately implies that greedy decoding is almost optimal under DDMC for LLM decoding problem when we view $u$ as the probability of LLM generating the sequence. This gives an evidence on why greedy decoding is unreasonably efficient for various domains despite its simplicity (Meister et al., 2020; Shi et al., 2024a; Team et al., 2023; Welleck et al., 2024), (see Song et al. (2024) for comprehensive evaluation of greedy decoding).

## 2. Problem Setup

We formally define the general problem setup, which will subsume tokenized linear bandit (TLB) and tokenized multi-armed bandit (TMAB) problems. Let $\mathcal{V}$ be the set of all possible tokens with $|\mathcal{V}| = n$. For any set $A$, let $A^*$ the collection of every possible finite sequences over $A$. Given a token sequence $\mathbf{y} \in \mathcal{V}^*$, we write $\mathbf{y}^{(i)} \in \mathcal{V}$ to denote its $i$-th token and $\mathbf{y}^{(i:j)} \in \mathcal{V}^*$ to denote the subsequence that contains every token from $i$-th token to $j$-th token including themselves for $j \geq i$ in $\mathbf{y}$.[6] We often write $\mathbf{y}^{(-i)}$ to denote the $i$-th token from the end of $\mathbf{y}$. There exists a special end-of-sequence EOS $\in \mathcal{V}$ token representing the end of a sequence. We call a token sequence $\mathbf{y}$ to be *complete* if its last token is EOS. Importantly, even though our sequence function could have nonzero value even without the EOS token at the end, we will *only* consider a class of algorithms that only ends with EOS token. Thus, throughout, we rewrite

$\mathcal{V}^*$ to denote the collection of every possible finite complete sequences. For $\mathbf{y}, \mathbf{z} \in \mathcal{V}^*$, we write $\mathbf{y} : \mathbf{z}$ to denote the token sequence that concatenates $\mathbf{y}$ and $\mathbf{z}$ sequentially. We often write $\mathbf{y} = \emptyset : \mathbf{y}$.

We call the process of sequential token selection in a single round the *decoding* algorithm, and denote the process of observing the feedback at the end of the decoding and learning the sequence function by *learning* phase. Essentially, one can view the decoding process as an *online problem* to select each token, and learning process as an *online learning problem* to estimate the objective function. We write $|\mathbf{y}|$ to denote the length (interchangeably, depth or level) of the sequence $\mathbf{y}$. We consider a class of decoding algorithms with fixed maximal *depth* $L$ that only outputs token sequence with length at most $L$. Equivalently, we assume $\mathcal{V}^*$ consists of token sequences with length up to $L$. We assume that the depth $L$ is known in advance to the DM.[7] The depth plays a crucial role in our analysis and any algorithm's performance would depend on $L$ that intrinsically captures the size of feasible token sequences. We refer to an algorithm's *level $l$* decision to denote its decision for the $l$-th token.

**Sequence utility and regret.** At each round $t \in [T]$, a user arrives and writes a query $x_t \in \mathcal{V}^*$. Then, a decision maker (DM) needs to select each token sequentially to construct a sequence $\mathbf{y}_t \in \mathcal{V}^*$ to answer the user's query, and the DM observes a reward $r_t(\mathbf{y}_t)$ when it finalizes the sequence selection process and displays $\mathbf{y}$. In particular, the DM irrevocably commits to each token to construct the token sequence and cannot revert any previous decision to change the sequence entirely. The reward is defined by two factors: (i) a sequence (utility) function $u_t : \mathcal{V}^* \to \mathbb{R}_{\geq 0}$ that maps a token sequence to a nonnegative real value and (ii) a random noise over it. Note here that $u_t$ can vary across the rounds, which captures the (possibly) different context provided to the DM given by the queries $x_t$ over the rounds.[8] Formally, the random reward $r_t(\mathbf{y}_t)$ for the sequence $\mathbf{y}$ can be represented as

$$r_t(\mathbf{y}_t) = u_t(\mathbf{y}_t) + \eta_t,$$

where $\eta_t$ is a random noise with $|\eta_t| \leq \sigma$.[9] The DM does not know the sequence function $u$ in advance, but can only indirectly access it through the reward $r_t(\mathbf{y}_t)$. The goal of the DM is to maximize the user utility over the rounds $T$, *i.e.*, $\sum_{t \in [T]} r_t(\mathbf{y}_t)$. Equivalently, the objective is to compete with the optimal complete sequence in hindsight: OPT =

---

[5] We also compare DDMC with smoothness assumption in BTS problem in Appendix C.

[6] We use boldface for sequences and non-boldface for a token.

[7] Practically, it can be prompted in the LLM or implicitly set in the LLM to ensure it does not generate an infinitely long output.

[8] An acute reader may easily observe that one cannot learn anything if $u_t$ does not have any structural assumption. Thus, Section 3 allows efficient learning by having a linear structure, and Section 4 allows efficient learning my having $u_t = u$.

[9] We assume bounded noise for simplicity, but it easily extends to bounded sub-Gaussian noise as in the bandit literature.

$\sum_{t=1}^{T} \text{OPT}_t$ where $\text{OPT}_t = \mathbb{E}\left[\max_{\mathbf{z} \in \mathcal{V}^*} r_t(\mathbf{z})\right]$. Then, the DM tries to minimize the expected cumulative (pseudo-) regret $\text{REG} = \sum_{t \in [T]} \text{REG}_t = \text{OPT} - \sum_{t \in [T]} \mathbb{E}\left[r_t(\mathbf{y}_t)\right]$.

**Monotonicity and single-level deviation.** Remark that we consider a class of algorithms that always output a complete sequence. In practice as well, once the sequence meets EOS, it is natural to stop the decoding. To capture such behavior, we impose the following assumptions.

**Assumption 2.1** (Monotonicity). *The utility function $u_t$ is monotone if (i) appending more* EOS *to a sequence that already has* EOS *at the end does not change the utility, and (ii) appending non-*EOS *token to a sequence that already has* EOS *at the end only decreases the utility.*

The monotonicity assumption is completely innocuous in a sense that appending EOS tokens to a complete sequence must not change the utility in practice as the user observes exactly the same output, and further any reasonable algorithm would submit the output once it meets EOS.

Further, let $\mathbf{a}_t$ be a sequence which is not optimal. If a sequence $\mathbf{a}_t$ deviates from the hindsight optimal sequence $\mathbf{o}_t$ firstly at level $i$, any practical algorithm should have small single-level error in a sense that the utility difference at the moment is almost ignorable as they differ only by a single token. Formally, we impose the following assumption.

**Assumption 2.2** (Small single-level deviation). *Let $\mathbf{a}$ be a sequence that deviates from the optimal sequence $\mathbf{o}$ firstly at level $i$. The utility function $u_t$ has single-level deviation (SLD) of $\varepsilon$ if $\left| u_t(\mathbf{o}^{(1:i)}) - u_t(\mathbf{a}^{(1:i)}) \right| \le \varepsilon$. Further, the utility function has small SLD (SSLD) if $\varepsilon = O(1/\sqrt{T})$.*

This assumption is apparently innocuous for large $i$ as adding a single token would almost incur no difference. Even if $i$ is small, one can expect that the utility difference will be still small as small $i$ indicates that the overall document only has a few tokens, which would not induce significantly different utility. For instance, one may consider truncating the token space to top-$k$ most probable ones, then SLD will be ignorable for small $i$ as any efficient LLM would only produce reasonable tokens as top-$k$ tokens.[10] Throughout, we will assume that the utility function is monotone and SSLD unless specified otherwise.

**LLM decoding problem.** Let us see the connection between our problem and *LLM decoding* problem and *LLM alignment* problem. Given an access to the logit probability of a LLM and a token sequence $\mathbf{y} \in \mathcal{V}^*$, suppose the sequence function $u_t$ denotes the LLM's probability $p$ of generating the output $\mathbf{y}$, *i.e.,* $u_t(\mathbf{y}) = p(\mathbf{y}) = \prod_{k=1}^{n} p(\mathbf{y}^{(k)} | \mathbf{y}^{(1:k-1)})$. Then, our objective can be framed

as an online learning problem to learn the LLM's logit probability in an efficient manner.

A special case of this problem is a scenario where the DM has access to the value oracle of $u$, *e.g.,* knows the probability distribution, in advance. In this case, the objective boils down to the sequence function maximization problem (Li & Milenkovic, 2017; Tschiatschek et al., 2017; Mitrovic et al., 2018; 2019) where duplicated insertion of the same element is allowed, or from the LLM's perspective, the *LLM decoding* problem of designing efficient decoding algorithms (Freitag & Al-Onaizan, 2017; Kulikov et al., 2018; Holtzman et al., 2019; Chen et al., 2024). Correspondingly, our main results have several implication in these problems, which will be elaborated in Section 5.2.

**LLM alignment problem.** Further, one can consider a variant of our setup, termed *LLM alignment* problem, where we have a (frozen) LLM that gives us a probability distribution $p(\mathbf{y})$, but this LLM is (partially) misaligned with our objective function $u$. In this case, one standard approach would be to use the probability distribution $p(\mathbf{y})$ as a baseline for the decoding process, while learning the misaligned objective function $u$. If the misalignment allows specific structure, *e.g.,* $u_t(x_t, \mathbf{y}) = (1 - \gamma)p(x_t, \mathbf{y}) + \gamma f(x_t, \mathbf{y})$[11] for some latent function $f$ capturing the misalignment, we indeed show that our results can be extended to efficiently handle such misalignment in Section 5.1.

## 3. Tokenized Linear Bandit

The general setting we formulated, however, does not allow any efficient learning if the utility function $u_t$ indeed does not have structural correlation across the rounds. One natural framework to tackle such fundamental obstacle is to impose the *linear realizability* assumption in the online contextual learning literature. To this end, we introduce the following notion of embedding function:

**Definition 3.1** (Embedding function). A decoding algorithm has access to the embedding function $e : \mathcal{V}^* \times \mathcal{V}^* \to \mathbb{R}^d$ maps a token sequence to $d$-dimensional real vector.[12]

**Assumption 3.2** (Linear realizability). *There exists $\theta \in \mathbb{R}^d$ and embedding function $e(\cdot)$ such that the utility function can be represented by $u_t(x_t, \mathbf{y}) = \langle \theta, e(x_t, \mathbf{y}) \rangle$ for every query $x_t \in \mathcal{V}^*$ and token sequence $\mathbf{y} \in \mathcal{V}^*$.*

The DM has a query access to the embedding function during the decoding process.[13] Then, the DM's objective

---

[10]This does largely not restrict the class of problem instances we deal with, as the utility difference onwards could be significantly large.

[11]In fact, setting $\gamma = 0$ reduces to the pure LLM decoding problem, whereas setting $\gamma = 1$ reduces to our general problem of learning the latent sequence function, or equivalently, learning a LLM's behavior.

[12]One can deem the embedding function as the semantic embedding that the LLM computes for $\mathbf{y}$ given a query $x_t$.

[13]Our algorithm calls the embedding function $O(nLT)$ time.

is to efficiently learn the latent parameter $\theta$ to maximize the utility. We assume that $\|\theta\| \leq 1$ and $\|e(\mathbf{z})\| \leq 1$.[14] We finally note that such linear approximation of the reward model often appear in the LLM alignment literature (Cen et al., 2024; Yang et al., 2024).[15]

### 3.1. Diminishing distance with more commons (DDMC)

On the other hand, even if the utility function is well-structured with the linear realizability assumption, the intrinsic intricacy of TLB's tokenized nature imparts a fundamental hardness in maximizing the reward. Intuitively, if a (monotone) sequence function could have arbitrary structures, one small mistakes in the first token selection may incur irreparable results in the following tokens, *e.g.,* imagine there exists only a single token $\tau$ which incurs nonnegative for $u(\tau : \mathbf{y})$ for any $\mathbf{y}$. If the first token itself does not give much information about the following levels, one cannot fundamentally identify an efficient path in hindsight.[16] This is captured by the following proposition:

**Proposition 3.3.** *For TLB problem, any algorithm suffers the worst-case regret of* $\Omega(T(1 - 1/2^{L-2}))$.

Thus, we here introduce a fairly natural structural assumption on the sequence function $u(\cdot)$ to avoid such pessimistic scenario, dubbed *diminishing distance with more commons*.

**Assumption 3.4** (DDMC). *A sequence function $u(\cdot)$ has diminishing distance with more commons (DDMC) if*

$$|u(x_t, \mathbf{y} : \tau) - u(x_t, \mathbf{z} : \tau)| \leq |u(x_t, \mathbf{y}) - u(x_t, \mathbf{z})|,$$

*for any two different finite sequences $\mathbf{y}, \mathbf{z}$ with the same length $|\mathbf{y}| = |\mathbf{z}|$ and any token $\tau \in \mathcal{V}$.* [17]

In other words, adding common tokens at the end of two sequences with the same length only decrease the utility gap between them.

DDMC shares very similar intuition from the widely adopted *submodularity* assumptions on set function (Korte et al., 2011), or recently sequence function (Mitrovic et al., 2019), which characterizes diminishing returns. However, they crucially differ and cannot be implied by each other, due to two key differences: (1) submodularity is about diminishing *return* (i.e., the difference between two function values) whereas DDMC is about diminishing *distance* (i.e.,

absolute value of the difference); (2) more subtly, DDMC is about the difference of two different sequences of the same length with an added common token, whereas submodularity is about the difference of two different sets that one includes another with an added common element. In some sense, DDMC is the complement of submodularity.

*Discussion 3.5.* From the practical perspective for LLM, this is innocuous since if two generated outputs shares common tokens suffix-wise, we can expect that they give the user similar experiences as long as we add more commons.

### 3.2. Excessive Optimism under the Face of Uncertainty

We present our algorithm, named *excessive optimism under the face of uncertainty* for tokenized linear bandits (EO-FUL), whose pseudocode is presented in Algorithm 1. It has *two optimistic* features. First, it optimistically decodes each token in a greedy manner, expecting that this would lead to a good utility in following decisions. Second, it optimistically estimates each token's utility by constructing a confidence ball $C_t$, and computes the best possible utility each token can gain from parameters in $C_t$.

Our construction of the confidence ball is analogous to the standard LinUCB (Li et al., 2010; Abbasi-Yadkori et al., 2011) algorithm for linear bandits.

Let $\mathbf{y}_t$ be the chosen sequence at round $t$ by Algorithm 1. We recursively define the following matrices for $t \in [T]$:

$$\Sigma_t = \Sigma_{t-1} + \mathbf{y}_t \mathbf{y}_t^\top,$$

and let $\Sigma_1 = \lambda I$ for $d \times d$ identity matrix $I$. Further, we define

$$\beta_t = \sigma^2 \left(2 + 4d \log\left(1 + \frac{tL}{d}\right) + 8 \log \frac{4}{\delta}\right).$$

For $t \geq 2$, the center of our confidence ball will be:[18]

$$\hat{\theta}_t = \Sigma_t^{-1} \sum_{i=1}^{t-1} r_t(x_t, \mathbf{y}_t)\mathbf{y}_t, \tag{3.1}$$

and $\hat{\theta}_1 = \vec{0}$, the $d$-dimensional zero vector.[19]

Now, our confidence ball $C_t$ to be used at round $t$ is defined as follows:

$$C_t = \{\vartheta : (\vartheta - \hat{\theta}_t)^\top \Sigma_t (\vartheta - \hat{\theta}_t) \leq \beta_t\}, \tag{3.2}$$

and $C_1 = \{\vec{0}\}$. We choose $\lambda = \Theta(1)$.

We further impose the following technical assumption:

---

[14]This is for ease of exposition to exclude cumbersome details, but all results would easily generalize under any bounded scenario.

[15]One might find this partly related to linear representation hypothesis (Park et al., 2023), stating that concepts are encoded as linear directions in the model's embedding space.

[16]In fact, due to the online learning nature of our problem, there always exists a small probability that any algorithm makes an error in the first level.

[17]In Appendix B, we provide a relaxed version of DDMC assumption and provide corresponding efficient algorithms.

[18]Remark that this is a solution of the ridge regression given the reward feedback.

[19]For efficient computation, one may use the rarely-switching version of OFUL algorithm in (Abbasi-Yadkori et al., 2011), instead of the standard OFUL.

---

**ALGORITHM 1:** EOFUL

**Input:** Token set $\mathcal{V}$, linear contextual bandit algorithm ALG

Initialize $\mathbf{y} \leftarrow \emptyset, \hat{\theta}_1 \leftarrow \vec{0}, C_1 = \{\hat{\theta}\}, \mathbf{y} = \emptyset$

**for** $t = 1, 2, \ldots, T$ **do**

    User arrives and submits query $x_t$

    **for** $k = 1, 2, \ldots, L - 1$ **do**

        Compute $\tau^* = \arg\max_{\tau \in \mathcal{V}, \theta \in C_t} \langle \theta, e(x_t, \mathbf{y} : \tau) \rangle$

        Set $\mathbf{y} \leftarrow \mathbf{y} : \tau$

        **if** $\tau =$ EOS **then** Break;

    Submit $\mathbf{y}$ and observe reward $r_t$

    Compute $\hat{\theta}_{t+1}$ by (3.1) and $C_{t+1}$ by (3.2)

---

**Assumption 3.6.** *For any complete sequence $\mathbf{y}$ with $|\mathbf{y}| = k$ and any subsequence $\mathbf{y}^{(1:l)}$ with $l \leq k$, there exists a $c \geq 0$ such that $\left\| \mathbf{y}^{(1:l)} \right\|_{\Sigma_t} \leq c \left\| \mathbf{y} \right\|_{\Sigma_t}$ in Mahalanobis norm.*[20]

That is, each entry of the subsequence of a complete sequence is not too larger than that of the complete sequence with respect to $\Sigma_t$. Both structural assumptions 3.2 and 3.6 are removed in Section 4 for TMAB.

*Discussion* 3.7. This assumption is innocuous in a sense that $c$ can be sufficiently large up to $o(\sqrt{T})$ to obtain sublinear regret. On the other hand, this is less innocuous since it depends on the algorithm's construction of $\Sigma_t$. In practice, one of the typical usage for LLM is to either (i) have similar queries based on their daily lives, or (ii) ask queries in already existing sessions (in-context) to expand the discussion. In these scenarios, we expect the queries and the responses thereby to be not too much different from each other.

*Discussion* 3.8. On the other hand, if $\mathbf{y}_i^{(1:l)}$ and $\mathbf{y}_t$ does not have any structural dependency, any algorithm would not be able to fundamentally learn certain coordinate of $\theta$. For instance, suppose any complete sequence's embedding only has nonzero values in the last $d/2$ dimension, and the first $d/2$ dimension for any incomplete sequence. Then, any algorithm could not learn the first $d/2$ dimension, implying that any internal node does not provide information on an efficient path to the complete sequence.

**Theorem 3.9.** *Impose assumption 3.6 and DDMC. With probability at least $1 - \delta$, EOFUL has regret $O(cL\sqrt{dT(\log T + \log(\frac{1}{\delta}))})$, and thus plugging $\delta = 1/T$ yields $\text{REG} = O(cL\sqrt{dT\log T})$.*

Overall, our proof proceeds as follows:

*Step 1 (Length equalization).* We first equalize $\mathbf{o}_t$ and $\mathbf{y}_t$ to have the same lengths in a way that it does not hurt any following analysis.

*Step 2 (Level-$k$ regret).* We define level $k$ regret as the regret occurred at depth $k$ of the decoding process that will play crucial role in the analysis to recursively transform the regret into upper level regrets.

*Step 3 (Fictitious extension).* Then we introduce a fictitious extension of $\mathbf{y}_t^{(1:k-1)}$ to $\mathbf{f}_t^{(1:k)} = \mathbf{y}_t^{(1:k-1)} : \mathbf{o}_t^{(k)}$, and show how the level-$k$ regret of the fictitious extension is upper bounded by level-($k$-1) regret at $\mathbf{y}_t^{(1:k-1)}$.

*Step 4 (Regret decomposition).* We relate the level-($k$-1) regret of the algorithm's choice $\mathbf{y}_t$ and level-$k$ regret of the fictitious extension using DDMC assumption, which upper bounds our algorithm's level-$k$ regret due to the optimistic choice of the algorithm. Then, we telescope over $k, \ldots, 1$ to express the entire regret as a function of level-1 regret plus some estimation errors occurred for each prefix of $\mathbf{y}_t$ during the decoding.

*Step 5 (Sum of squares regret)* Finally, we compute the function of level-1 regret and the estimation error by following the standard sum of squares regret bound analysis of linear contextual bandit algorithm, where readers familiar with the standard linear bandit analysis may easily follow.

## 4. Tokenized Multi-armed Bandit

The linear realizability assumption is essential for the contextual setting to derive any efficient algorithm. A natural follow-up question is whether one could obtain an efficient algorithm without linear realizability, if the context does not change and the utility function $u_t = u$ remains the same over the rounds.[21] In this case, the DM's problem can be seen as a variant of the standard multi-armed bandit problem where it needs to make an (online) decision for the next token irrevocably to construct a complete sequence. Thus, we dub this problem *tokenized multi-armed bandit* (TMAB).

We first argue that even in this simple setting, our problem admits a pessimistic lower bound when the sequence function does not have any structural assumption beyond monotonicity. Then, we again introduce DDMC and present an algorithm with regret $O(nLT^{2/3}(\log T)^{1/3})$. We present an intimate connection between TMAB and the bandit-based tree search (BTS) problem (Coquelin & Munos, 2007) in Appendix C.

**Exponential lower bound.** If the query is fixed, our objective boils down to efficiently learn the latent utility function $u$. Without any structural assumption on it, however, one can easily see that it is equivalent to learn the reward distribution of every subsequence $\mathbf{y} \in \mathcal{V}_L^*$. This is because each subsequence $\mathbf{y}$ can be deemed as a single arm in the standard multi-armed bandit problem which admits regret lower bound of $\Omega(\sqrt{KT})$ where $K$ is the number of arms. This is formally stated as follows:

**Proposition 4.1.** *For TMAB problem, any algorithm has worst-case regret lower bound of $\Omega(\min(\sqrt{n^L T}, T))$.*

---

[20]Mahalanobis norm is defined by $\|x\|_A = \sqrt{x^\top A x}$.

[21]We omit the context $x_t$ in the inputs of $u$ since the context remains the same.

**DDMC and GreedyETC.** The pessimistic exponential dependency on depth $L$ naturally motivates the use of DDMC Assumption 3.4. Unlike the TLB problem, however, there exists no global latent parameter $\theta$ that we could use throughout the decoding process, but they can genuinely arbitrary beyond DDMC (and monotonicity). Thus we use a rather direct approach to explore each level sufficiently, and commit to the seemingly optimal token at each level then move on to the next level.

---

**ALGORITHM 2:** GreedyETC

---

**Input:** Token set $\mathcal{V}$, exploration parameter $N$
Initialize $\mathbf{y} \leftarrow \emptyset$
**for** $k = 1, 2, \ldots, L$ **do**
    **if** $|y| = L - 1$ **then**
        $\mathbf{y} \leftarrow \mathbf{y} : \texttt{EOS}$
        Break
    **for** $t \in \mathcal{V}$ **do**
        Submit $\mathbf{y} : t : \texttt{EOS}$ for $N$ times and observe rewards
        Set $\bar{r}(\mathbf{y} : t)$ be the averaged cumulative rewards
    Select $\tau^* = \arg\max_{t \in \mathcal{V}} \bar{r}(\mathbf{y} : \tau)$
    Set $\mathbf{y} \leftarrow \mathbf{y} : \tau^*$
    **if** $\tau^* = \texttt{EOS}$ **then**
        Break
For the rest of the rounds, submit $\mathbf{y}$

---

We prove that under (a slight variant of formally presented in) DDMC assumption, a natural combination of greedy decoding with exploration-then-commit (ETC) style of algorithm, as presented in Algorithm 2, yields sublinear regret w.r.t. $T$ with linear dependency on $d$:

**Theorem 4.2.** *Under Assumption 2.1 and E.6, Algorithm 2 achieves regret $O(nLT^{2/3}(\log T)^{1/3})$.*

One subtle part is that, if one naively construct the *clean* event that the rewards for each sequence are concentrated to its expectation and take union bound over all of them, this essentially gives the regret of $\Omega(n^L)$, which is undesirable. Instead, we only take the union bound along the path that our algorithm is selected, which enables us to recover the linear dependency on $L$. Then, we obtain a recursive inequality relating the regret at level $k$ with that at level $k - 1$ by considering fictitious extension as defined in the proof of Theorem 3.9, and conclude that the regret remains small with high probability.

# 5. Applications

## 5.1. LLM alignment

One interesting application of our results is the LLM alignment problem. In this problem, we want to generate aligned outputs given access to a frozen LLM that cannot be further fine-tuned. Consider a LLM parameterized by a token distribution $p : \mathcal{V}^* \times \mathcal{V}^* \to [0, 1]$ that maps any token sequence given a user's query to a probability that the LLM generates this sequence. The user's utility given

token sequence $\mathbf{y}$ is misaligned from the LLM such that $u(x_t, \mathbf{y}) = \gamma v(p(x_t, \mathbf{y})) + (1 - \gamma)f(x_t, \mathbf{y})$, where $v$ is a monotone function that maps probability to reward and $f$ is the latent function to be learned that marginalizes the misaligned part of the LLM.

Such convex combination between multiple rewards typically appear in the LLM alignment literature (Rame et al., 2024; Shahriar et al., 2024; Jang et al., 2023; Shi et al., 2024b), when one wants to align with multiple human preferences by well combining multiple models or parameters. Our utility function as the convex combination over $v$ and $f$ can also be interpreted in a similar spirit. One typical example for the function $v$ would be the log probability or perplexity obtained from the frozen LLM, or more explicit user utility such as classification accuracy or factuality (Lin et al., 2022). We assume the DM knows the probability to utility mapping function $v$.

If we assume $f$ is linearly realizable, *i.e.,* there exists an embedding function $e : \mathcal{V}^* \times \mathcal{V}^* \to \mathbb{R}^d$ such that $f(\cdot) = \langle \theta, e(x_t, \mathbf{y}_t) \rangle$, we can reduce the problem of learning misaligned function $f$ to the TLB problem. Formally, we construct $\theta' \in \mathbb{R}^{d+1}$ as a vector that concatenates $\theta$ and $1$ in $d + 1$-th dimension after multiplying each by $1 - \gamma$ and $\gamma$ respectively, *i.e.,* $\theta' = [(1 - \gamma) \cdot \theta : \gamma \cdot 1]$. Then we can rewrite the utility as

$$u(x_t, \mathbf{y}) = \langle \theta, (1 - \gamma)e(x_t, \mathbf{y}_t) \rangle + \gamma v(p(x_t, \mathbf{y}))$$
$$= \langle \theta', [e(x_t, \mathbf{y}) : v(p(x_t, \mathbf{y}))] \rangle .$$

This is exactly the feedback structure in our TLB problem, where the DM observes $e(x_t, \mathbf{y})$ and $v(p(x_t, \mathbf{y}))$ to construct the concatenated vector and tries to learn $\theta'$. Correspondingly, EOFUL enjoys the same regret bound in Theorem 3.9 for this problem.[22]

## 5.2. LLM decoding

Our results and analysis therein have a few interesting implications in (i) the sequence function maximization problem (Li & Milenkovic, 2017; Tschiatschek et al., 2017; Mitrovic et al., 2018; Alaei et al., 2021) and (ii) the problem of designing efficient decoding algorithms for text generation in language model (Freitag & Al-Onaizan, 2017; Kulikov et al., 2018; Holtzman et al., 2019; Chen et al., 2024), if we consider a special case when the function $u$ is known to the DM and consider a static version with a single round.

First, this is exactly a sequence function maximization problem to maximize $u$ given query access to a value oracle that

---

[22]Note that our formulation differs from standard KL-divergence based problem (Yang et al., 2024). One might integrate the KL divergence in our problem too by redefining the function over the distribution rather than a fixed sequence, however, it's not clear to formulate in the language of multi-armed bandit problem. This remains an interesting future direction.

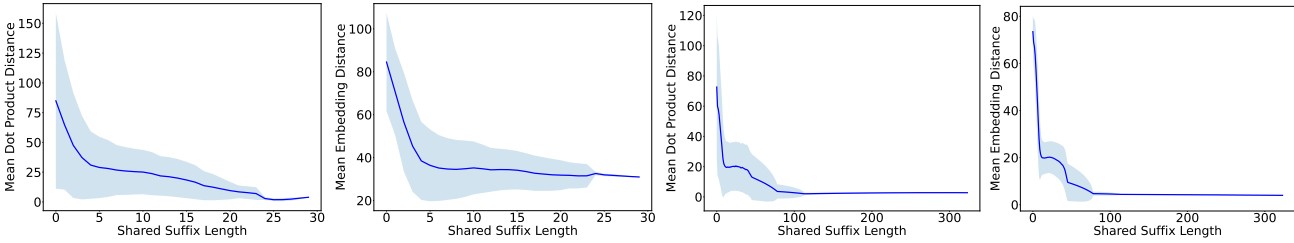

Figure 1. The first two show distance gap under TruthfulQA and the next two is under HH-RLHF, with distance function $d_1$ and $d_2$.

outputs $u(x_t, \mathbf{y})$ given a sequence $\mathbf{y}$. One common objective here is to obtain an approximate algorithm with polynomial query complexity. If $u$ satisfies a notion of monotonicity and sequence-submodularity (Alaei et al., 2021), it is known that the greedy algorithm has $1 - 1/e$ approximation.[23] On the other hand, our result implies that greedy algorithm is almost optimal under DDMC assumption, formally stated as follows:[24]

**Theorem 5.1.** *For any monotone sequence function, greedy decoding is almost optimal under DDMC in a sense that it is $(1 - \varepsilon)$-approximation where $\varepsilon$ is the SLD parameter.*

From the LLM perspective, this result has one interesting corollary. Given a fixed query $x$, consider a LLM parameterized by $p : \mathcal{V}^* \times \mathcal{V}^* \to [0, 1]$. Then, Theorem 5.1 implies that if $p$ is monotone DDMC function, the greedy decoding can exactly output the token sequence that maximizes the logit probability. This supports the unreasonable effectiveness of the greedy decoding, despite its simplicity, justifying why it works well and widely adopted in the literature (Meister et al., 2020; Shi et al., 2024a; Song et al., 2024), *e.g.,* in Google's Gemini report (Team et al., 2023), even though it is only known as an approximate maximum-a-posterior decoding algorithm (Welleck et al., 2024).

On the other hand, greedy decoding is often dominated by sampling methods such as nucleus sampling (Holtzman et al., 2020) or self-consistency (Wang et al., 2022) for various tasks, *e.g.,* (Song et al., 2024) find that vanilla multinomial sampling can be better than greedy for open-ended creative generation. We conjecture that such phenomenon largely depends on how the sequence function generated by LLM, *i.e.,* the logit probabilities, would look like and thereby the type of the tasks. Investigating other regimes that another decoding algorithm provably outperforms greedy remains an intriguing open question.

---

[23]It also admits the APX-hardness with $1 - 1/e$ by the hardness of submodular set-function maximization from Maximum Coverage, unless P=NP.

[24]We remark that our setup is slightly different from standard sequence maximization problems as we allow the selection of the same element over different position, and the special EOS token governs the end of the sequence by naturally encoding it in the monotonicity of the utility function.

## 6. Experiments

We finally provide our experimental results.

### 6.1. Validating DDMC

The first is on validation of DDMC assumption. Further experimental results are presented in Appendix D.

**Setup.** For the embedding function $e$, we obtain the embedding from Llama3-8B-Instruct model (AI@Meta, 2024). For the token sequences, to validate our DDMC in real-world datasets, we use TruthfulQA dataset (Lin et al., 2022) and HH-RLHF dataset (Bai et al., 2022),[25] by concatenating each query and answer pair.

For each item in the dataset, we group every (query:answer) pair by number of common tokens in its suffix. For instance, image the following two token sequences:

$$\mathbf{x} = (a, b, c, d, e, f), \ \mathbf{y} = (a, x, y, d, e, f).$$

Then, we can group each subsequence as the following:

- 0 common suffix: $(\mathbf{x}^{(1:2)}, \mathbf{y}^{(1:2)}), (\mathbf{x}^{(1:3)}, \mathbf{y}^{(1:3)})$
- 1 common suffix: $(\mathbf{x}^{(1:1)}, \mathbf{y}^{(1:1)}), (\mathbf{x}^{(1:4)}, \mathbf{y}^{(1:4)})$
- 2 common suffix: $(\mathbf{x}^{(1:5)}, \mathbf{y}^{(1:5)})$
- 3 common suffix: $(\mathbf{x}^{(1:6)}, \mathbf{y}^{(1:6)})$

We evaluate each distance $d(\mathbf{x}^{(1:k)}, \mathbf{y}^{(1:k)})$ in each group and obtain the average and variance.

For the distance functions, we evaluate two candidates:

$$d_1(x, y) = \langle \theta, \mathbf{x} - \mathbf{y} \rangle$$
$$d_2(x, y) = \|\mathbf{x} - \mathbf{y}\|_2,$$

where $\theta = \vec{1}_{4096}$, *i.e.,* 4096-dimensional one vector. The first distance function $d_1$ is consistent with our DDMC

---

[25]Both datasets are standard for LLM alignment, where TruthfulQA validates truthfulness and hallucination and HH-RLHF evaluates helpfulness and harmlessness. In HH-RLHF, data are organized in (prompt:response) format. For simplicity, we reuse "query" to refer to "prompt" and "answer" to refer to "response".

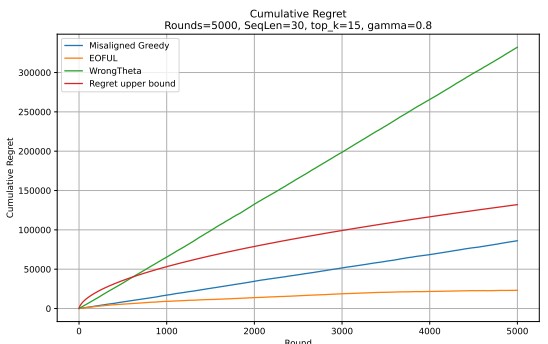 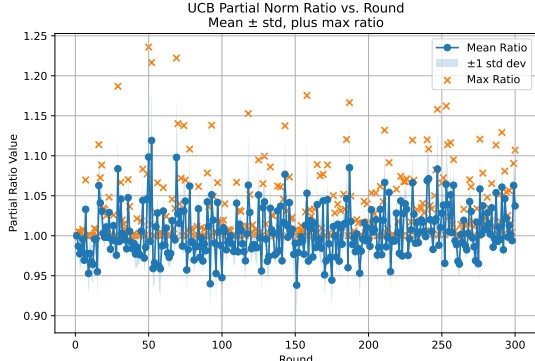

*Figure 2.* The first figure compares the regret of algorithms. The second figure represents the ratio $\rho_t^l = \left\| \mathbf{y}^{(1:l)} \right\|_{\Sigma_t} / \left\| \mathbf{y} \right\|_{\Sigma_t}$ over the rounds. Max Ratio and Mean Ratio denote the maximum and average ratio, respectively, for $l \in [L]$ at each round.

assumption, and it in fact can be viewed as a $\ell_1$ distance between two vectors. Thus, the second distance function $d_2$ can be viewed as a generalization of $d_1$ into $\ell_2$ distance.

**Result.** The first two figures in Figure 1 refers to the TruthfulQA, and the next two denotes the HH-RLHF. Each experiment is presented for distance function $d_1$ and then $d_2$. Overall, we observe a tendency to have decreasing distance gap as the number of common tokens in suffices increases. This holds for both the distance function $d_1$ and $d_2$. Interestingly, the decreasing curvature tends to be stronger when the number of common tokens are rather small. Further, the decreasing gap seems to be more drastic under HH-RLHF, suggesting that different tasks might possess different structures in their sequence function, *i.e.,* logit probability structure. Overall, despite verified in an average sense, this justifies the DDMC assumption, which we believe could be largely useful for following works.

### 6.2. Performance of EOFUL

We numerically validate the performance of EOFUL using synthetic data under the LLM alignment scenario presented in Section 5.1. In particular, we set the length of each sentence to be $L = 30$, and truncate top-15 tokens for every algorithm for efficiency. Further, we set $\gamma = 0.8$, $\theta = (0.5, 0.5, \ldots, 0.5)$. Dteails in the query generation can be found in Appendix D. We compare three reasonable benchmarks: (i) theoretical regret upper bound (*under-scaled* by 0.1 to make the plot more visible), (ii) WrongTheta that uses wrongly estimated $\theta' = (-0.5, -0.5, \ldots, -0.5)$ and choose a token that maximizes the resulting convex combination, and (iii) Misaligned greedy that simply performs greedy decoding with respect to the LLM's probability. As observed in the first part of Figure 2, we observe that EOFUL effectively achieves sublinear regret. Given that the regret upper bound is under-scaled,the actual performance

may be much better than that of the theoretical guarantees.

In practice, one could blend some standard alignment techniques with our learning based approaches. For instance, one can run standard alignment algorithms in the earlier rounds as EOFUL could suffer significant errors in the beginning. In the mean time one can operate EOFUL in backgrounds to learn the confidence region, and then adopt a combination of them in later rounds. This will be way more practical than naively running EOFUL particularly if the effect of coldstart in EOFUL, *e.g.,* apparently misleading or grammatically incorrect outcomes, could significantly degrade the user experience.

**Validating Assumption 3.6.** Further, in the same experimental setup, we validate Assumption 3.6 for our EOFUL. Notably, we observe that the ratio is universally upper bounded above by 1.25, even after taking maximum over every token position. Recalling that such upper bound scales the regret upper bound in a linear manner, this would only blow up the regret to be 1.25 times that of the standard linear contextual bandit, let alone the linear dependency on $L$.

## 7. Conclusion

We introduce tokenized variants of linear and multi-armed bandit problems, where a decision maker needs to maximize cumulative rewards which can be represented as a randomly perturbed sequence function by selecting each token in a sequential manner. We introduce a novel but natural assumption called diminishing distance with more commons (DDMC), and show that both problems admit efficient algorithms with regret linear on the depth of the maximal sequence and sublinear on the time horizon. We provide several applications and implications on LLM alignment problem as well as decoding problem, and validate our assumptions and performance of our algorithms through synthetic and real-world datasets.

## Acknowledgement

Part of the work is done while Suho Shin is visiting Computer Science at the University of Chicago. This work is partially supported by DARPA QuICC, ONR MURI 2024 award on Algorithms, Learning, and Game Theory, Army-Research Laboratory (ARL) grant W911NF2410052, NSF AF:Small grants 2218678, 2114269, 2347322. Haifeng Xu is supported by the AI2050 program at Schmidt Sciences (Grant G-24-66104), Army Research Office Award W911NF23-1-0030 and NSF Award CCF-2303372.

## Impact Statement

This paper aims to advance the understanding of the multi-armed bandit problem, LLM alignment and decoding. There could be several potential societal consequences of our work, none of which we feel must be specifically highlighted here.

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

# A. Related work

**Multi-armed bandit and variants.** Dating back to (Lai & Robbins, 1985), multi-armed bandits and its linear bandit variant have been central to modern recommender system (Auer, 2002; Li et al., 2010; Abbasi-Yadkori et al., 2011). There exists a few works investigating its variant to tree search model (Coquelin & Munos, 2007), analyzing efficient algorithms under certain structural assumption, *e.g.,* smoothness of tree.[26]

**Decoding-time alignment.** There are a line of works that use heuristic methods for decoding-time alignment using external modules that guide the decoding process. (Josifoski et al., 2022) suggested aligning the LLM in the decoding time to align the likelihood and utility based on some external information. (Mudgal et al., 2023) present a token level Markov decision process (MDP) framework called controlled decoding using a separate prefix scorer rule that is trained to learn a value function from rewards. (Huang et al., 2024) present a decoding scheme as a heuristic-guided search process to align various preferences. (Chakraborty et al., 2024) propose a transfer decoding to transfer-learn the unknown $Q$ function in the token level MDP, and analyze the upper bound on the gap between the optimal policy and their algorithm. Meanwhile, our decoding-time alignment problem is solely framed as a bandit problem not MDP.

**Foundation of alignment.** Recent works have provided some theoretical foundations on LLM alignment. (Yang et al., 2024) derived a closed-form characterization of the optimal KL-constrained reinforcement learning (RL) alignment problem, and conclude that the famous best-of-$N$ policy and KL-constrained RL is asymptotically equivalent. (Beirami et al., 2024) analyze the upper bound on the KL divergence of best-of-$N$ policy. We remark that our LLM alignment problem is formulated in a largely different manner (see Section 5.1), thus less relevant to these literature.

**Foundation of decoding.** (Basu et al., 2020) analyze the perplexity of several sampling methods assuming that the token probabilities follow a Zipf distribution. (Finlayson et al., 2023) provided a theoretical justification on using truncation to avoid the excessive probability assignment on unreliable tail, empirically observed by (Holtzman et al., 2019). Recently, Chen et al. (2024) propose a decoding problem as a two-player game, and provide near-optimal strategies that encompass greedy decoding and some of its variants. Our LLM decoding problem is formulated as a sequence function maximization problem to maximize the posterior probability given a user query, which is largely different from above.

# B. $k$-DDMC and Lookahead Decoding

Interestingly, we can relax the DDMC assumption and obtain analogous results by replacing the greedy decoding part with the *lookahead* decoding. In $k$-lookahead decoding, it generates all possible combination of the following $k$ tokens, and select the token sequence that maximizes the utility (probability). This allows us to recover the fictitious extension argument used in both the analysis of TLB and TMAB.

First, $k$-DDMC is defined as follows:

**Assumption B.1** ($k$-DDMC). *A sequence function $u(\cdot)$ is $k$-DDMC if*

$$| u(x_t, \boldsymbol{y} : \boldsymbol{a}) - u(x_t, \boldsymbol{z} : \boldsymbol{a}) | \leq | u(x_t, \boldsymbol{y}) - u(x_t, \boldsymbol{z}) |,$$

*for any two finite sequences $\boldsymbol{x}, \boldsymbol{y}$ with the same length $|\boldsymbol{x}| = |\boldsymbol{y}|$, any sequence $\boldsymbol{a} \in \mathcal{V}^*$ with $|\boldsymbol{a}| = k$, and any query $x_t$ such that LHS is nonnegative.*

Correspondingly, we can modify our algorithms as presented in Algorithm 3 and Algorithm 4.

Now we given an outline of how one could extend our analysis for EOFUL and GreedyETC. Notice that the key part for the regret analysis is to compare an algorithm's choice and optimal algorithm's choice $\mathbf{a}_t^{(1:i)}$ and $\mathbf{o}_t^{(1:i)}$, respectively, up to $i$-th token at round $t$ by considering a fictitious extension $\mathbf{f} = \mathbf{a}_t^{(1:i)} : \mathbf{o}_t^{(i+1)}$. With $k$-DDMC, we extend this analysis by considering alternative fictitious extension such that $\mathbf{f} = \mathbf{a}_t^{(1:i)} : \mathbf{o}_t^{(i+1:i+k)}$. With such extension, until level $j \leq k$, our algorithm is guaranteed to decode the (empirically) optimal token sequence given the estimates.

---

[26]More detailed discussion can be found in Appendix C.

---

**ALGORITHM 3:** $k$-Lookahead-EOFUL
**Input:** Token set $\mathcal{V}$, linear contextual bandit algorithm ALG
Initialize $\mathbf{y} \leftarrow \emptyset$, $\hat{\theta}_1 \leftarrow \vec{0}$, $C_1 = \{\hat{\theta}\}$, $\mathbf{y} = \emptyset$
**for** $t = 1, 2, \ldots, T$ **do**
  User arrives and submits query $x_t$
  **for** $k = 1, 2, \ldots, L - 1$ **do**
    Compute $\mathbf{z}^* = \arg\max_{\mathbf{z} \in \mathcal{V}^*, |\mathbf{z}| = k, \theta \in C_t} \langle \theta, e(x_t, \mathbf{y} : \mathbf{z}) \rangle$
    Set $\mathbf{y} \leftarrow \mathbf{y} : \mathbf{z}^*$
    **if** $(z^*)^{(-1)} = $ EOS **then** Break;
  Submit $\mathbf{y}$ and observe reward $r_t$
  Compute $\hat{\theta}_{t+1}$ by (3.1) and $C_{t+1}$ by (3.2)

---

**ALGORITHM 4:** $k$-LookaheadETC
**Input:** Token set $\mathcal{V}$, exploration parameter $N$
Initialize $\mathbf{y} \leftarrow \emptyset$
**for** $k = 1, 2, \ldots, L$ **do**
  **if** $|\mathbf{y}| = L - 1$ **then**
    $\mathbf{y} \leftarrow \mathbf{y} : $ EOS
    Break
  **for** $\mathbf{z} \in \mathcal{V}^*$ *with* $|\mathbf{z}| = k$ **do**
    Submit $\mathbf{y} : t : $ EOS for $N$ times and observe rewards
    Set $\bar{r}(\mathbf{y} : \mathbf{z})$ be the averaged cumulative rewards
  Select $\mathbf{z}^* = \arg\max_{\mathbf{z} \in \mathcal{V}^*, |\mathbf{z}| = k} \bar{r}(\mathbf{y} : \mathbf{z})$
  Set $\mathbf{y} \leftarrow \mathbf{y} : \mathbf{z}^*$
  **if** $(z^*)^{(-1)} = $ EOS **then**
    Break
For the rest of the rounds, submit $\mathbf{y}$

---

Then, for level $j > k$ such that $j = i + k$, we have:

$$\mathbf{o}_t^{(1:j)} - \mathbf{a}_t^{(1:j)} = \mathbf{o}_t^{(1:i+k)} - \mathbf{a}_t^{(1:i+k)} = \mathbf{o}_t^{(1:i+k)} - \mathbf{f} + \mathbf{f} - \mathbf{a}_t^{(1:i+k)}$$
$$\leq \left| \mathbf{o}_t^{(1:i)} - \mathbf{a}_t^{(1:i)} \right| + \mathbf{f} - \mathbf{a}_t^{(1:i+k)} \qquad \text{(By } k\text{-DDMC)}$$
$$\leq \left| \mathbf{o}_t^{(1:i)} - \mathbf{a}_t^{(1:i)} \right|, \qquad \text{(By optimistic choice of algorithms)}$$

thereby connecting $j$-level regret to $j - k$-level regret. With telescoping arguments and following our analysis for EOFUL and GreedyETC, one can prove that $k$-Lookahead-EOFUL and $k$-LookaheadETC can obtain regret sublinear in $T$ and linear in $L$. We leave the detailed analysis to the readers.

## C. Connection to Bandit Tree Search problem

In bandit tree search (BTS) problem by (Coquelin & Munos, 2007), there exists a binary tree with depth $d$. Each leaf node $i$ is assigned a random variable $X_i$ with bounded support $[0, 1]$ and expectation $\mu_i$. At each round $t \in [T]$, the DM chooses a leaf node $I_t$ and observe the random reward $X_{I_t}$. The DM, however, is unknown about the tree structure and needs to explore it. Likewise, the DM irrevocably selects each child in the tree until it reaches a leaf node. A static and decision version of this problem is to decide whether there exists a leaf node $l$ whose utility function $v(l) \geq A$ for some $A \geq 0$. Likewise, a static and decision version of TMAB is to decide whether there exists a subsequence $\mathbf{y}$ with $u(\mathbf{y}) \geq A$ for some $A \geq 0$ in polynomial time. We will see that static decision versions of these problems can be reduced from each other. Illustrative examples of such reductions are provided in Figure 3.

The following theorem formally argues that one can reduce BTS to TMAB:

**Theorem C.1.** *An instance of static and decision version of BTS can be reduced to an instance of static and decision version of TMAB and monotone utility function.*[27]

*Proof.* Let $\mathcal{L}$ be the set of leads and $d$ be the depth of the tree in BTS. For each leaf node $l \in \mathcal{L}$, we construct a corresponding

---

[27]The reduction can easily be extended to $k$-ary tree version of BTS.

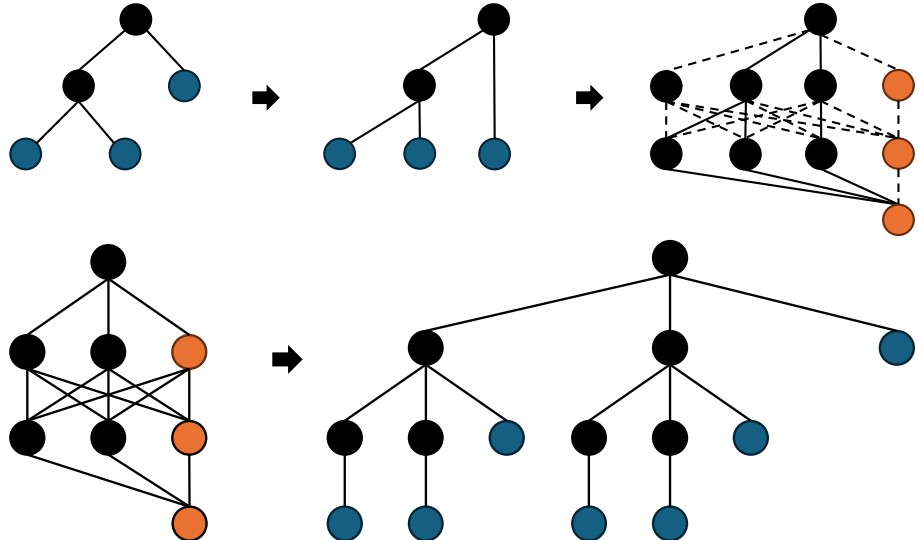

*Figure 3.* Illustration of reductions. The above figure is an example of reduction from BTS to TMAB, and the below is from TMAB to generalized BTS (with $k$-ary tree). In both figures, blue nodes represent leaf nodes and orange node denotes EOS token. Dotted edge in the rightmost graph implies that the sequence contains such edge would have utility zero. Thus, only the sequences without dotted lines have (potentially) nonzero utility corresponding to the utility at the leaf node in the original instance of BTS. In the below figure, each path to EOS in the instance of TMAB becomes a leaf node in the instance of generalized BTS.

token $\tau(l) \in \mathcal{V}$ and add EOS token, *i.e.,* $\mathcal{V} = \{\tau(l) : l \in \mathcal{L}\} \cup \{\text{EOS}\}$. Let $v(\cdot)$ be the value function for BTS that maps leaf to nonnegative real values. We write $d(x)$ to denote the depth of the any (possibly internal) node $x$ in the instance of BTS. Let $A_l$ be the set of nodes in depth $l$ in BTS. We arbitrarily define an injection $\sigma_l$ from $A_l$ to $\mathcal{V} \setminus \{\text{EOS}\}$. We say that $\tau \in \mathcal{V}$ is *covered* at level $l$ if there exists a node $x$ with depth $l$ in BTS such that $\sigma_l(x) = \tau$.

Now, we construct sequence function $u : \mathcal{V}^* \to \mathbb{R}_{\geq 0}$ as follows: for every $\mathbf{y} \in \mathcal{V}^*$

1. If $\mathbf{y}^{(-1)} \neq 0$, then $u(\mathbf{y}) = 0$.

2. If $\mathbf{y}^{(-1)} \neq 0$ and if every $\mathbf{y}^{(k)}$ for $k \in [|\mathbf{y} - 1|]$ is covered, then $u(\mathbf{y}) = v(\sigma_l^{-1}(\mathbf{y}^{(-2)}))$.

3. If $\mathbf{y}^{(-i)} = \text{EOS}$ for $i = 1, 2, \ldots, k$ for $k \geq 2$ and $\mathbf{y}^{(-k)} \neq \text{EOS}$, then $u(\mathbf{y}) = u(\mathbf{y}^{(1:|\mathbf{y}|-(k-1))})$.

It is straightforward to check that No instance of the static and decision version of BTS enforces No instance to that of TMAB, since no sequence would have utility larger than or equal to $A$. Consider Yes instance of the static and decision version of BTS and let $l$ be the leaf node with $v(l) \geq A$. Now we define $\mathbf{y}$ to satisfy:

1. $|\mathbf{y}| = d(l) + 1$.

2. $\mathbf{y}^{(k)} = \sigma_k(l)$ for every $k \in [d(l)]$.

3. $\mathbf{y}^{(d(l)+1)} = \text{EOS}$.

Then, due to our construction of the utility function above, we have $u(\mathbf{y}) = v(l) \geq A$, finishiing the reduction.

Now it remains to prove that the constructed utility function is monotonicity as defined in Assumption 2.1. This naturally follows by step 3 in the construction of $u(\cdot)$, since adding more EOS does not change the utility.

$\square$

We remark that one might think that TMAB is not a search problem unlike BTS since we know the set of feasible sequences in advance, but it can indeed be viewed as a search problem since the ultimate goal is to find a node with larger, nonzero, utility.

**Theorem C.2.** *An instance of static and decision version of TMAB can be reduced to an instance of static and decision version of TS.*

*Proof.* We sort the token set $\mathcal{V}$ and index by $\sigma : \mathcal{V} \to \{1, 2, \ldots, n\}$ where such that $\sigma(\text{EOS}) = n$. Given an instance of BTS problem with $\mathcal{V}$ we construct $n$-ary tree. For depth $d$ of the tree, there will be $n - 1$ NORMAL nodes who have $n$ children, and one END node with no children. We will arbitrarily assign index $i \in [n - 1]$ to each normal node and $n$ to the END node.

Consider any complete sequence $\mathbf{y}$ with length $l$. Then, we can select a corresponding leaf node $l$ such that along the path $p$ to $l$, the node in depth $d$ in path $p$ has index $\sigma(\mathbf{y}^{(d)})$. Since $\mathbf{y}$ is complete, the corresponding leaf node $l$ is indeed a leaf since its last index is $\sigma(\mathbf{y}^{(-1)}) = n$, which is END node that does not have any children. Thus, we assign utility $v(l) = u(\mathbf{y})$. Then, the reduction easily follows, and we leave the details to the readers. $\square$

*Discussion* C.3. Accordingly, one natural question is what it means by the binary tree in BTS having DDMC structure. As noted in the proof and Figure 3, the DDMC assumption in TS means that each node has exactly the same set of children, which corresponds to the given token set. Then, one can encode the DDMC assumption in TS by considering each node as the corresponding specific token.

**DDMC assumption in BTS, smoothness in TMAB.** Given the close connection between two problems, one might wonder what the DDMC assumption means in BTS. If we consider $k$-ary tree version of BTS, then we can imagine an ordering between children for each node, let the index be $1, 2, \ldots, n$. Then, in BTS, our DDMC assumption asserts that each two nodes at the same depth append the children with the same index $i$, the resulting utility gap only decreases compared to the utility gap between the parent nodes.

On the other hand, (Coquelin & Munos, 2007) presents a rather immediate and strong assumption on the tree, named smoothness, stated as follows:

**Assumption C.4** (Smoothness). *Given an instance of BTS, for any depth $d < L$, there exists $\delta_d > 0$ such that for at least one optimal node $i$ of depth $d$ (that leads to the optimal leaf node), for all leaves $j$ in the subtree of $i$, it follows that*

$$u(l^*) - u(u_j) \leq \delta_d,$$

*where $l^*$ denotes the optimal leaf node.*

That is, it upper bounds the utility gap at depth $d$ by the parameter $\delta_d$, and provide a regret upper bound parameterized by such $\delta_d$. This could analogously be applied in our TMAB problem, by upper bounding the utility gap between any sequence of length $l$ and the subsequence of length $l$ that leads to the optimal subsequence by some parameters $\delta_l$. On the other hand, our DDMC assumption does not introduce any such upper bound on the utility gap directly, but rather imposes a minor structural relation between sequences with length difference $1$. Thus, we believe that our DDMC assumption can also play a crucial role in the BTS problem, given that it imposes certain structures on the tree implicitly, and the empirical evidence captured in Section 6.

# D. More Experiments and Omitted Details

**Details of experiments in Section 6.** For the queries for regret comparison and assumption 3.6 validation, we randomly generate the prompt each round based on a manually specified list of templates and a list of user's interests.

Specifically, we use the following template for prompts to generate queries.

[ "I work as a software engineer in New York and I love interest. Any advice on balancing both?", "I'm looking for ways to improve my skills in interest without burning out at my tech job. Any tips?", "Could you recommend some resources or places in NYC to enjoy interest after work?", "What are some practical time-management strategies to fit in interest while coding full-time?", "Any suggestions on how to meet new people in the city who also enjoy interest?", "Can you give me a one-sentence recommendation for exploring interest around New York?", "I often hang out with coworkers who share my interest in interest. How do we plan something fun after work?", "As someone who loves interest and coding, how can I unwind effectively?", "What are some must-try experiences in interest near Manhattan on weekends?", "Could you suggest a beginner-friendly way to dive into interest?", "Any advice on juggling a programming career and daily sessions of interest to stay productive?", "What is a good way to combine my love of interest and my passion for software engineering?", "I'd like to find new ways to level up my interest skills during lunch breaks. Suggestions?", "Any personal organization tips for balancing a full-time software job with interest and social life?", "In your opinion, what's a fun project or event in NYC that merges interest with technology?", "I love going out with friends to enjoy interest, but I'm also studying new programming languages. How do I manage both?", "Could you recommend an app or a platform that helps me track progress on interest?", "Any recommended communities in New York for people who share my enthusiasm for interest?", "What's an underrated aspect of interest that helps relieve stress from my coding job?", "I'm trying to step out of my comfort zone with interest. Got any beginner challenges or ideas?" ]

The following is a list of user's interest.

[ "tennis", "dog training", "video gaming", "breweries and nightlife", "coding projects", "tech meetups", "coffee shops", "gym and workouts", "weekend getaways", "music concerts", "NYC events", "board game nights", "watching sports with friends", "side hustle ideas", "improving my coding skills", "finding time for hobbies", "balancing health and tech", "staying updated with new frameworks", "apartment-friendly dog breeds", "managing stress after work", ]

Formally, we randomly generate 1000 queries. For each query generation, we randomly sample a user's interest and template and then format the prompt. The set of generated queries are used throughout the experiments by randomly sampling from the set.

### D.1. More experiments

**Regret comparison.** It turns out that depending on the parameters and the randomness of the experiments, EOFUL often suffer slower convergence as presented in Figure 4. Slower convergence might be the artifact of smaller top-$k$ truncation of tokens, as EOFUL does not have much freedom to explore according to what it truly desires. There could be a trade-off in setting top-$k$ truncations. For instance, if we set large $k$, this gives the opportunity of EOFUL to learn exactly as it desires, which might ideally lead to fast exploration and learning of the latent parameter. On the other hand, decreasing $k$ could filter out apparently wrong answers in EOFUL or any other learning-based approaches, *i.e.,* it neutralizes the impact of cold start. One can further adaptively set top-$k$ to fine-tune the algorithms in practice.

**Sample response.** Here we provide a few sample responses from the empirical evaluation with synthetic dataset presented in Section 6. Recall that $\theta$ is set to be $(1/2, 1/2, \ldots, 1/2)$ artificially, so the optimal sequence here with respect to our linear utility function may not capture the most preferred answer by the user.

First, below are sample responses and corresponding utilities and regrets in earlier rounds $t = 1$ and $t = 2$.

---

**Earlier rounds samples**

1. Query: I work as a software engineer in New York and I love managing stress after work. Any advice on balancing both?

2. Optimal greedy's utility: $-1.36$

    (a) Response: I'm more than happy to provide some helpful tips, my fellow software engineering and wellness-seekering New York-based friend! Here are a few strategies ...

3. EOFUL's utility: $-26.64$

    (a) Response: Ah-ha! Balancing work and stress-mangling (or stress-mangling-then- stress-avoid-then- stress-avoid-then ...

4. WrongTheta's utility: $-34.98$

    (a) Response: Stress after work is common especially after dealing with long code lines, tight deadlines, and high expectations. Balancing work and self-care is ...

5. Misaligned greedy's utility: $-9.90$

    (a) Response: Hi there! I'm happy to help. Balancing work and personal life can be challenging, especially in a fast-paced city like New York. Here...

---

**Earlier rounds samples**

1. Query: Any advice on juggling a programming career and daily sessions of board game nights to stay productive?

2. Optimal greedy's utility: $3.30$

    (a) Response: I'd be happy to help with balancing a programming career and leisure activities, which also contribute positively to personal well-being and social connections, which ultimately can ...

3. EOFUL's utility: $-17.48$

    (a) Response: Juggling a programming career and daily board game nights can indeed be challenging, but it's definitely doable and even potentially beneficial in some unexpected ways Here ...

4. WrongTheta's utility: $-41.49$

    (a) Response: Ahah Juggling is all about balance! Firstly, prioritize and set clear boundaries for yourself. Allocate specific time slots for work and board game ...

5. Misaligned greedy's utility: $-17.78$

    (a) Response: Ahaha, that sounds like a fun challenge! Juggling a programming career and board game nights can be done, but it requires some planning and discipline ...

---

As one may see in the sample response of the EOFUL, the responses are often grammatically incorrect and write redundant words repeatedly. Also, utility seems to be significantly smaller than that of the benchmark (optimal greedy), as learning requires more rounds due to high dimensional nature of the embedding space as well as the complexity of the sequence function learning.

Note also that WrongTheta provides vacuously wrong answers, though it is not expected in hindsight. For instance, if the wrongly set $\theta$ captures the true preference in hindsight, it should be the case that it produces something meaningful. Thus, this specific algorithm yielding apparently incorrect outcomes could be simply an artifact of randomness in experiments. Further, as expected, statically setting $\theta$ as a fixed estimator cannot learn the true $\theta$ over the rounds, showcasing the superiority of EOFUL approach.

On the other hand, below are sample response at the end of the rounds, $t = 9999$ and $t = 10000$

---

**Later rounds samples**

1. Query: I'd like to find new ways to level up my staying updated with new frameworks skills during lunch breaks. Suggestions?

2. Optimal greedy's utility: $-2.99$

   (a) Response: Here are a few ideas for leveraging your 15- or 30-minute lunch breaks to learn something new about your favorite programming frameworks or technologies: ...

3. EOFUL's utility: $-1.57$

   (a) Response: Here are a few ideas for leveraging your 12:00 - 1:00 PM lunch breaks to learn about new frameworks and skills: ...

4. WrongTheta's utility: $56.36$

   (a) Response: Ah-ha! Stuck in front of a screen? Don '92 u2019 u20ac u2029 u20ac u2019 ...

5. Misaligned greedy's utility: $-11.06$

   (a) Response: That's a great goal! Here are some ideas to help you level up your skills during lunch breaks: 1. **Code Challenges**: Websites like Hacker ...

---

**Later rounds samples**

1. Query: As someone who loves improving my coding skills and coding, how can I unwind effectively?

2. Optimal greedy's utility: $22.87$

   (a) Response: Great concern, my fellow coding enthusiast! I'm more than happy to help you unwind effectively. Here are some strategies that many developers find helpful:

3. EOFUL's utility: $28.34$

   (a) Response: Great concern, my fellow coding enthusiast! I'm more than happy to share some ways that help many fellow developers, and perhaps, even the best coding skills ...

4. WrongTheta's utility: $-36.90$

   (a) Response: Unwinding after a gruelling coding marathon or after a long day of
   n
   n
   n staring at code? Unwinding ...

5. Misaligned greedy's utility: $2.68$

   (a) Response: Unwinding is essential for a coder's mental and physical well-being. Here are some effective ways to help you relax and recharge: ...

---

Perhaps surprisingly, though expected from theoretical results in some perspectives, EOFUL is mimicking the response of the optimal greedy very effectively. It almost outputs more than 10 same words from the beginning. Once it deviates, however, due to the nature of the sequence function generation, it is doomed to be very different onwards, which is fairly expected consequence. Further, one can observe that EOFUL's utility often outperform the optimal greedy's one, due to the online nature of the sequence function maximization. This also indicates that greedy decoding is not necessarily optimal for some circumstances.

**More experiments on DDMC.** In Figure 5, we further validate DDMC assumptions on two more datasets: AdvBench

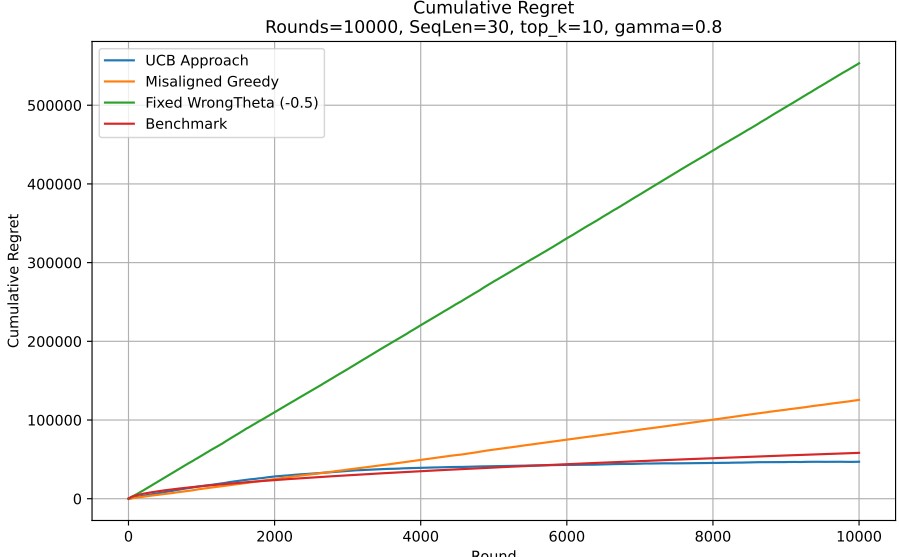

*Figure 4.* $T = 10000$ and $L = 30$. We truncate to use only top 10 tokens and set $\gamma = 0.8$.

(standard jailbreak benchmark) (Zou et al., 2023) and just-eval-instruct (AI2, 2024) that contains many prompts of various tasks. Further, we verified DDMC assumptions beyond our linear utility function. We considered three more functions: $l_1, l_3, l_4$ distance, beyond the inner product distance and $l_2$ distance presented in Section 6. In all these scenarios, we observe a tendency that appending more common tokens decreases the utility gaps, validating that DDMC, at least in expectation or its relaxed version, is practical in many scenarios.

# E. Proofs

## E.1. Proof of Proposition 3.3

*Proof of Proposition 3.3.* Let $\mathcal{L}$ be the set of sequences with length $L$ so that $|\mathcal{L}| = 2^L$ and we arbitrarily index the elements in $\mathcal{L}$ and write $\mathcal{L} = \{\mathbf{z}_1, \mathbf{z}_2, \ldots, \mathbf{z}_{2^L}\}$. Let $o$ be the one-hot vector who has one only at the first coordinate and otherwise zero, and set $\theta = o$. Consider a choice of the query sequence $(x_t)_{t \in [T]}$ and the embedding function such that $e(x_t, \mathbf{y})$ has a unique subsequence $\mathbf{y} \in \mathcal{L}$ such that $e(x_t, \mathbf{y}) = o$, and another unique subsequence $\mathbf{z} \in \mathcal{L}$ such that $\mathbf{z}^{(1:L-1)} = \mathbf{y}^{(1:L-1)}$ and $e(x_t, \mathbf{z}) = (1 - \varepsilon, 0, 0, \ldots, 0)$ for SLD $\varepsilon$, and otherwise $e(x_t, \mathbf{y}) = 0$ for every other $\mathbf{y} \in \mathcal{V}^*$. Further, assume that the sequence $(x_t)_{t \in [T]}$ ensures that such unique $\mathbf{y} \in \mathcal{L}$ is selected uniformly at random over $\mathcal{L}$. Let the chosen unique such vectors at round $t$ be $\mathbf{y}_t$ and $\mathbf{z}_t$. Note that this is indeed possible by having large enough candidates for $x_t$ whose cardinality scales as $\Theta(T)$ and for each $x_t$ it independently and uniform randomly selects the unique such $\mathbf{y} \in \mathcal{L}$ with $e(x_t, \mathbf{y}) = 0$ and correspondingly set $\mathbf{z}_t$. It is straightforward to check that the corresponding utility function satisfies Assumption 2.1 and 2.2. Observe that such instance is in fact a randomization over deterministic instances each of which deterministically choose $\mathbf{y}_t \in \mathcal{L}$ for each $t \in [T]$. Thus, by Yao's minimax principle (Yao, 1977), it suffices to consider a class of deterministic algorithms.

In this instance, there always exists a sequence $\mathbf{y} \in \mathcal{L}$ such that $e(x_t, \mathbf{y}) = o$, and thus the optimal decoding reward in hindsight is to always select $\mathbf{y} \in \mathcal{L}$ with $e(x_t, \mathbf{y}) = o$, which induces the cumulative reward of $T \cdot \langle o, o \rangle = T$. On the other hand, since the choice of $\mathbf{y}_t$ is completely independent from the previously chosen sequences and realization of random variables therein, any deterministic algorithm's probability to choose either of $\mathbf{y}_t$ or $\mathbf{z}_t$ is at most the random guess over $2^{L-2}$ elements until it reaches the subtree with children $\mathbf{y}$ and $\mathbf{z}$, resulting in the cumulative reward of $T \cdot \langle o, o \rangle / 2^{L-2} = T/2^{L-2}$. Note here that this is $2^{L-2}$ instead of $2^{L-2}$ since once the algorithm selects $\mathbf{y}_t^{(1:L-2)}$, it reward is upper bounded by an algorithm that deterministically chooses $\mathbf{y}_t^{(L-1)}$. Thus, the regret is at least $T(1 - 1/2^{L-2})$. $\square$

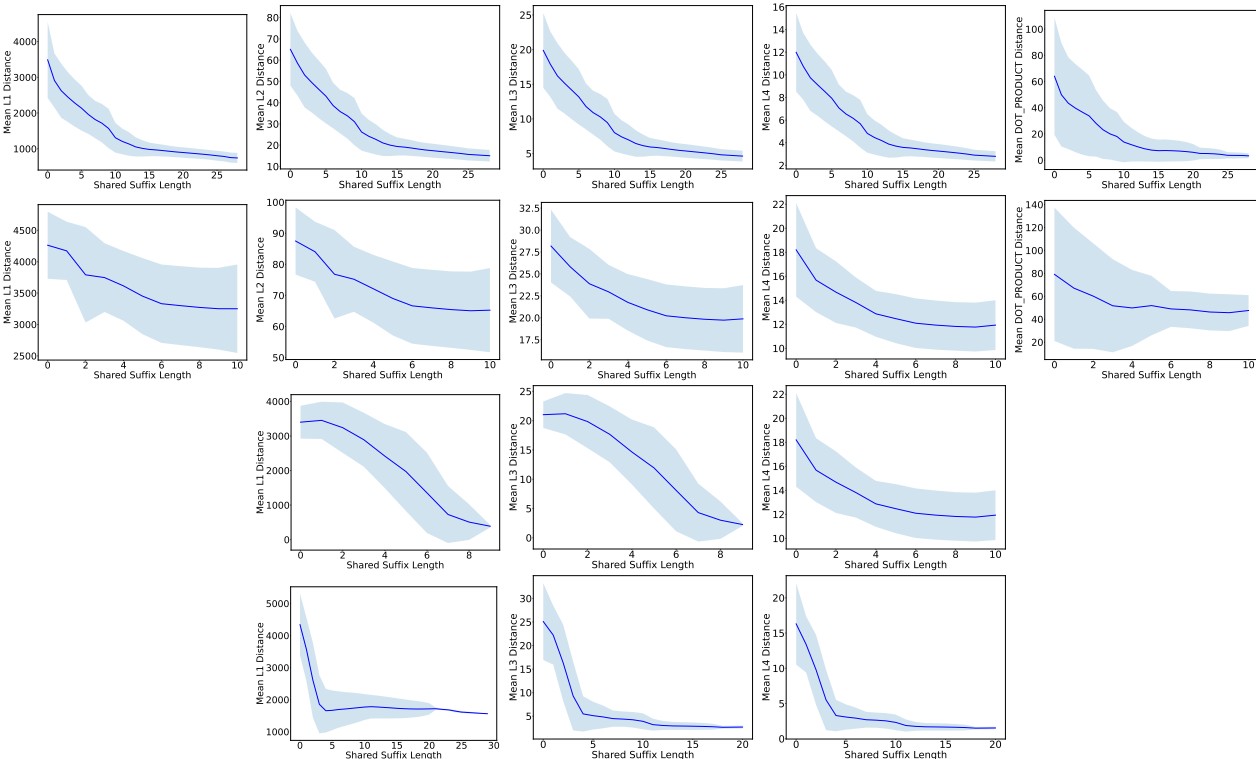

*Figure 5.* Each column represents each dataset. From top to bottom, each column represents AdvBench, HH-RHLF, Just-Eval, and TruthfulQA.

## E.2. Proof of Theorem 3.9

We here provide the proof of Theorem 3.9. Statement and proofs of the required technical lemmas are deferred to the end of this section.

*Proof of Theorem 3.9.* Throughout, we assume that `Clean` holds defined in Lemma E.1, which occurs with probability at least $\delta$.

Fix a round $t$, and let $\tilde{\theta}_t^{(k)} \in C_t$ be the optimistic estimator chosen at round $t$ for $k$-th token. Let $\mathbf{o}_t$ be the optimal sequence chosen by the optimal algorithm in hindsight at round $t$. For ease of exposition, in representing utility given sequence $\mathbf{y} \in \mathcal{V}^*$ and query $x_t$, we abuse $\langle \theta, \mathbf{y} \rangle$ to denote $\langle \theta, e(x_t, \mathbf{y}) \rangle$ unless confusion arises, *i.e.,* consider $\mathbf{y}$ to be the resulting embedded vector $e(x_t, \mathbf{y}) \in \mathbb{R}^d$.

*(Length equalization)* We will first define a level-$k$ regret that computes the difference of utility between the $k$-prefix of $\mathbf{o}_t$ and $\mathbf{y}_t$. However, this is not well-defined for every $k \in \max(|\mathbf{o}_t|, |\mathbf{y}_t|)$ if $\mathbf{o}_t$ and $\mathbf{y}_t$ have different lengths. Thus, we consider the *length equalization* operation on $\mathbf{o}_t$ and $\mathbf{y}_t$ that constructs extensions $\tilde{\mathbf{o}}$ and $\tilde{\mathbf{a}}$ of $\mathbf{o}$ and $\mathbf{a}$ respectively to have the same length in what follows.

We have two cases: (i) if $|\mathbf{a}| > |\mathbf{o}|$, then we append multiple `EOS` tokens at the end of $\mathbf{o}$ and construct $\tilde{\mathbf{o}}$ so that $|\tilde{\mathbf{o}}| = |\mathbf{a}|$, and (ii) otherwise $|\mathbf{a}| < |\mathbf{o}|$, then we do the same operation for $\mathbf{a}$ to construct $\tilde{\mathbf{a}}$ with $|\tilde{\mathbf{a}}| = |\tilde{\mathbf{o}}|$. Then, we extend both sequences $\tilde{\mathbf{a}}$ and $\tilde{\mathbf{o}}$ to exactly have the length $L$ by appending `EOS`. By Assumption 2.1, we have $u(\mathbf{o}) = u(\tilde{\mathbf{o}})$ and $u(\mathbf{a}) = u(\tilde{\mathbf{a}})$. Thus, we can safely deal with the extended sequences with the same length as the regret remains the same. For ease of exposition, we override $\mathbf{a}_t \leftarrow \tilde{\mathbf{a}}_t$ and $\mathbf{o}_t \leftarrow \tilde{\mathbf{o}}_t$ for every $t \in [T]$.

*(Level-$k$ regret and fictitious extension)* Now, we can formally define level-$k$ regret as follows:

$$\text{REG}_t^{(k)} = \left\langle \theta, \mathbf{o}_t^{(1:k)} \right\rangle - \left\langle \theta, \mathbf{y}_t^{(1:k)} \right\rangle.$$

Now we observe the connection between the level-$k$ regret with different $k$. For $k = 2, \ldots, L$, we consider an one-step *fictitious extension* of $\mathbf{y}_t^{(1:k-1)}$ that appends the optimal sequence $\mathbf{o}_t$'s $k$-th token to $\mathbf{y}_t$'s ($k$-1)-th prefix, *i.e.*,

$$\mathbf{f}_t^{(1:k)} = \mathbf{y}_t^{(1:k-1)} : \mathbf{o}_t^{(k)}$$

Notice that $\mathbf{o}_t^{(1:k)}$ and $\mathbf{f}_t^{(1:k)}$ shares the same common suffix $\mathbf{o}^{(k)}$.

Thus, by DDMC assumption, behold:

$$\left\langle \theta, \mathbf{o}_t^{(1:k)} \right\rangle - \left\langle \theta, \mathbf{f}_t^{(1:k)} \right\rangle \leq \left\langle \theta, \mathbf{o}_t^{(1:k-1)} \right\rangle - \left\langle \theta, \mathbf{f}_t^{(1:k-1)} \right\rangle$$

$$= \left\langle \theta, \mathbf{o}_t^{(1:k-1)} \right\rangle - \left\langle \theta, \mathbf{y}_t^{(1:k-1)} \right\rangle \tag{E.1}$$

$$= \mathrm{REG}_t^{(k-1)} \tag{E.2}$$

*(Regret decomposition)* Now, for $k \geq 2$, the level-$k$ regret can be decomposed as follows:

$$\mathrm{REG}_t^{(k)} = \left\langle \theta, \mathbf{o}_t^{(1:k)} \right\rangle - \left\langle \theta, \mathbf{y}_t^{(1:k)} \right\rangle$$

$$= (a) + (b) + (c),$$

where

$$(a) = \left\langle \theta, \mathbf{o}_t^{(1:k)} \right\rangle - \left\langle \theta, \mathbf{f}_t^{(1:k)} \right\rangle$$

$$(b) = \left\langle \theta, \mathbf{f}^{(1:k)} \right\rangle - \left\langle \tilde{\theta}_t^{(k)}, \mathbf{y}_t^{(1:k)} \right\rangle$$

$$(c) = \left\langle \tilde{\theta}_t^{(k)}, \mathbf{y}_t^{(1:k)} \right\rangle - \left\langle \theta, \mathbf{y}_t^{(1:k)} \right\rangle$$

Assume, first, that $\mathrm{REG}_t^{(k)} \geq 0$ for every $k$. By (E.2), we immediately have $(a) \leq \left| \mathrm{REG}_t^{(k-1)} \right|$.

Since our algorithm chooses $\mathbf{y}_t^{(k)}$ instead of $\mathbf{o}_t^{(k)}$ given $\mathbf{y}_t^{(1:k-1)}$ for $k$-th token, by the optimistic choice we have $(b) \leq 0$.[28]

Let us denote the term in $(c)$ by $W_t^{(k)}$ for $k \in [M]$.

Thus, we have

$$\mathrm{REG}_t^{(k)} \leq \left| \mathrm{REG}_t^{(k-1)} \right| + W_t^{(k)}.$$

Let $i$ be the first level that $\mathbf{y}_t$ deviates from $\mathbf{o}_t$. Telescoping over $k$ from $i$ to $L$:

$$\mathrm{REG}_t \leq \sum_{k=i}^{L} \left| W_t^{(k)} \right| + \left| \mathrm{REG}_t^{(i)} \right|.$$

Due to SSLD assumption, we have $\mathrm{REG}_t^{(i)} \leq O(1/\sqrt{T})$, so we can effectively ignore this term as it contributes at most $\sqrt{T}$, and thus we abuse

$$\mathrm{REG}_t \leq \sum_{k=1}^{L} |W_t^{(k)}|.$$

Define $w_t(\mathbf{x}) = \sqrt{\mathbf{x}^\top \Sigma_t^{-1} \mathbf{x}}$ for $\mathbf{x} \in \mathbb{R}^d$. Analogous to the standard regret analysis, we eventually use the sum of square regret arguments.

---

[28]We note that this argument remains the same regardless of the length equalization process above. For instance, even if $\mathbf{y}_t^{(k-1)} = \mathtt{EOS}$ and thus $\mathbf{y}_t^{(k)} = \mathtt{EOS}$, by Assumption 2.1, appending $\mathtt{EOS}$ is the greedy choice.

Now we invoke Lemma E.3 to obtain upper bounds for $W_t^{(k)}$.

For $k = 2, 3, \ldots, L$, notice that:

$$
\begin{aligned}
\left| W_t^{(k)} \right| &= \left| \left\langle \tilde{\theta}_t^{(k)}, \mathbf{y}_t^{(1:k)} \right\rangle - \left\langle \theta, \mathbf{y}_t^{(1:k)} \right\rangle \right| \\
&= \left| \left\langle \tilde{\theta}_t^{(k)}, \mathbf{y}_t^{(1:k)} \right\rangle - \left\langle \hat{\theta}_t, \mathbf{y}_t^{(1:k)} \right\rangle \right| + \left| \left\langle \hat{\theta}_t, \mathbf{y}_t^{(1:k)} \right\rangle - \left\langle \theta, \mathbf{y}_t^{(1:k)} \right\rangle \right| \\
&\leq 2\sqrt{\beta_t} w_t(\mathbf{y}_t^{(1:k)}) \\
&\leq 2\sqrt{\beta_t} \min(w(\mathbf{y}_t^{(1:k)}), 1),
\end{aligned}
$$

where the first inequality holds since the regret is at most 2 for any round and the second inequality follows from Lemma E.3 since both $\tilde{\theta}_t^{(1)}, \theta \in C_t$ under Clean. Combining two, we have

$$
\text{REG}_t \leq 2\sqrt{\beta_t} \left( \sum_{k=1}^{L} \min(w(\mathbf{y}_t^{(1:k)}), 1) \right).
$$

Now, by Assumption 3.6, we have

$$
w(\mathbf{y}_t^{(1:k)}) = \sqrt{(\mathbf{y}_t^{(1:k)})^\top \Sigma_t^{-1} \mathbf{y}_t^{(1:k)}} \leq c\sqrt{\mathbf{y}_t^\top \Sigma_t^{-1} \mathbf{y}_t} = cw(\mathbf{y}_t).
$$

Hence, we have

$$
\text{REG}_t \leq 2\sqrt{\beta_t}(1 + cL) \min(w_t(\mathbf{y}_t), 1).
$$

*(Sum of squares regret)* Squaring:

$$
\text{REG}_t^2 \leq 4\beta_t(1 + cL)^2 \min(w_t(\mathbf{y}_t)^2, 1),
$$

and

$$
\begin{aligned}
\sum_{t=1}^{T} \text{REG}_t^2 &\leq \sum_{t=1}^{T} 4\beta_t(1 + cL)^2 \min(w_t(\mathbf{y}_t)^2, 1) \\
&\leq 4\beta_T(1 + cL)^2 \sum_{t=1}^{T} \min(w_t(\mathbf{y}_t)^2, 1) \\
&\leq 4\beta_T(1 + cL)^2 \sum_{t=1}^{T} \ln(1 + w_t(\mathbf{y}_t)^2),
\end{aligned}
$$

where the last inequality uses the fact that for any $x \in [0, 1]$, $\ln(1 + x) \geq x/(1 + x) \geq x/2$, thus when $w_t(\mathbf{y}_t)^2 \leq 1$, we have $w_t(\mathbf{y}_t)^2 \leq 2\log(1 + w_t(\mathbf{y}_t)^2)$ and otherwise if $w_t(\mathbf{y}_t)^2 > 1$, we have

$$
4\beta_t = \frac{4}{\log 2}\beta_t \log 2 \leq \frac{4}{\log 2}\beta_t \log(1 + w_t(\mathbf{y}_t)^2).
$$

Then, we have

$$
\begin{aligned}
\sum_{t=1}^{T} \text{REG}_t^2 &\leq 4\beta_T(1 + cL)^2 \sum_{t=1}^{T} \ln(1 + w(\mathbf{y}_t)^2) \\
&\leq 4\beta_T(1 + cL)^2 d \ln\left(1 + \frac{T}{d\lambda}\right) \qquad \text{(By Lemma E.4 and E.5)}
\end{aligned}
$$

Finally, by Cauchy-Schwartz inequality,

$$\sum_{t=1}^{T} \text{REG}_t \leq \sqrt{T \cdot \sum_{t=1}^{T} \text{REG}_t^2}$$

$$\leq \sqrt{4T(1+cL)^2 \beta_T d \ln\left(1 + \frac{T}{d\lambda}\right)}$$

$$= O(cL\sqrt{dT \ln T}), \qquad\qquad \text{(Since } \beta_T = O(\lambda)\text{)}$$

$\square$

Now we provide the statements of lemmata. We begin with the standard concentration argument stating that $\theta \in C_t$ with high probability:

**Lemma E.1** (Confidence bound). *Define clean event:*

$$\texttt{Clean} = \{\theta \in C_t, \forall t \in [T]\}.$$

*Then,* $\Pr[\texttt{Clean}] \geq 1 - \delta$.

**Lemma E.2** (Self-normalized bound for martingale (Abbasi-Yadkori et al., 2011)). *Let $\{F_t\}_{t=0}^{\infty}$ be a filtration. Let $\{\eta_t\}_{t=1}^{\infty}$ be a real-valued stochastic process such that $\eta_t$ is $F_t$-measurable and $\eta_t$ is conditionally $R$-sub-Gaussian for some $R \geq 0$. Let $\{X_t\}_{t=1}^{\infty}$ be an $\mathbb{R}^d$-valued stochastic process such that $X_t$ is $F_{t-1}$-measurable. Assume that $V$ is a $d \times d$ positive-definite matrix. For any $t \geq 0$, define:*

$$\bar{V}_t = V + \sum_{s=1}^{t} X_s X_s^{\top}, \; S_t = \sum_{s=1}^{t} \eta_s X_s.$$

*Then for any $\delta > 0$, with probability at least $1 - \delta$, for all $t \geq 0$,*

$$\|S_t\|_{\bar{V}_t^{-1}}^2 \leq 2R^2 \log\left(\frac{\det(\bar{V}_t^{1/2} \det(V)^{1/2})}{\delta}\right).$$

For $\mathbf{z} \in \mathbb{R}^d$, define $w_t(\mathbf{z}) = \sqrt{\mathbf{z}^{\top} \Sigma_t \mathbf{z}}$. We then need the following several lemmata to proceed the sum of squares regret bound:

**Lemma E.3.** *For any $z \in \mathbb{R}^d$ and $\vartheta \in C_t$, we have*

$$\left\langle \vartheta - \hat{\theta}_t, z \right\rangle \leq 2\sqrt{\beta_t} w_t(z) \qquad\qquad (\text{E.3})$$

**Lemma E.4.** *The following equation holds:*

$$\det(\Sigma_T) = \det(\Sigma_1) \prod_{t=1}^{T} (1 + w_t(\mathbf{y}_t)^2).$$

**Lemma E.5.** *We have:*

$$\log \frac{\det(\Sigma_T)}{\det(\Sigma_1)} = d \log\left(1 + \frac{T}{d\lambda}\right).$$

The proofs are mostly similar to the standard linear bandit analysis (Abbasi-Yadkori et al., 2011), but is tailored to our setting. Thus, for completeness, we provide the proofs of some of the lemmata:

*Proof of Lemma E.1.* Note that the reward at round $t$ is $r_t = \langle \theta, e(x, \mathbf{y}_t) \rangle + \eta_t$. For concise exposition, we abuse the notation and write $\mathbf{y}_t = e(x, \mathbf{y}_t)$.

$$\hat{\theta}_t - \theta = \Sigma_t^{-1} \sum_{i=1}^{t} r_i \mathbf{y}_i - \theta$$

$$= \Sigma_t^{-1} \sum_{i=1}^{t} \mathbf{y}_i (\langle \theta, \mathbf{y}_i \rangle + \eta_t) - \theta$$

$$= \Sigma_t^{-1} (\mathbf{y}_i \mathbf{y}_i^\top) \theta - \theta + \Sigma_t^{-1} \sum_{i=1}^{t} \eta_i \mathbf{y}_i$$

$$= -\lambda \Sigma_t^{-1} \theta + \Sigma_t^{-1} \sum_{i=1}^{t} \eta_i \mathbf{y}_i.$$

Notice that

$$\sqrt{(\hat{\theta}_t - \theta) \Sigma_t (\hat{\theta}_t - \theta)^\top} = \left\| \Sigma_t^{1/2} (\hat{\theta}_t - \theta) \right\|$$

Expanding:

$$\left\| \Sigma_t^{1/2} (\hat{\theta}_t - \theta) \right\| \leq \left\| \lambda \Sigma_t^{-1/2} \theta \right\| + \left\| \Sigma_t^{-1/2} \sum_{i=1}^{t} \eta_i \mathbf{y}_i \right\|$$

$$\leq \sqrt{\lambda} \left\| \theta \right\| + \sqrt{2 \log \frac{\det(\Sigma_t) \det(\Sigma_1)^{-1}}{\delta_t}},$$

where the last inequality uses $\Sigma_t^{-1} \leq 1/\lambda$ due to the construction.

Note first that $\hat{\theta}_1 = \vec{0} \in C_1$ surely. For $t \geq 2$, we set $\delta_t = 3/(\pi^2 t^2) \cdot 2\delta$. Then, using union bound, one can deduce that

$$\Pr\left[ \theta \in C_t, \forall t \in [T] \right] \geq 1 - \delta.$$

$\square$

*Proof of Lemma E.3.* By Cauchy-Schwarz inequality and the fact that $\vartheta \in C_t$, we obtain the following series of inequality:

$$\left| (\vartheta - \hat{\theta}_t)^\top z \right| = \left| (\vartheta - \hat{\theta}_t)^\top \Sigma_t^{1/2} \Sigma_t^{-1/2} z \right|$$

$$= \left| (\Sigma_t^{1/2} (\vartheta - \hat{\theta}_t))^\top \Sigma_t^{-1/2} z \right|$$

$$\leq \left\| \Sigma_t^{1/2} (\vartheta - \hat{\theta}_t) \right\| \left\| \Sigma_t^{-1/2} z \right\| \quad \text{(Cauchy Schwarz inequality)}$$

$$= \left\| \Sigma_t^{1/2} (\vartheta - \hat{\theta}_t) \right\| \sqrt{z^\top \Sigma_t^{-1} z}$$

$$\leq \sqrt{\beta_t z^\top \Sigma_t^{-1} z}. \quad \text{(Since } \vartheta \in C_t\text{)}$$

$\square$

*Proof of Lemma E.4.* By construction of $\Sigma_t$, we have

$$\det(\Sigma_{t+1}) = \det(\Sigma_t + \mathbf{y}_t \mathbf{y}_t^\top)$$

$$= \det \left( \Sigma_t^{1/2} \left( I + \Sigma_t^{-1/2} \mathbf{y}_t \mathbf{y}_t^\top \right) \Sigma_t^{1/2} \right)$$

$$= \det(\Sigma_t) \det \left( I + \Sigma_t^{-1/2} \mathbf{y}_t \left( \Sigma_t^{-1/2} \mathbf{y}_t \right)^\top \right)$$

$$= \det(\Sigma_t) \det \left( I + v_t v_t^\top \right),$$

where we write $v_t = \Sigma_t^{-1/2} \mathbf{y}_t$. Observe that $v_t^\top v_t = w_t(\mathbf{y}_t)^2$. Further,

$$(I + v_t v_t^\top)v_t = v_t + v_t(v_t^\top v_t) = (1 + w_t(\mathbf{z}_t)^2)v_t,$$

implying that $(1 + w_t(\mathbf{y}_t)^2)$ is an eigenvalue of $I + v_t v_t^\top$. Since $v_t v_t^\top$ is a rank one matrix, all other eigenvalues of $I + v_t v_t^\top$ are one. Thus,

$$\det(\Sigma_{t+1}) = \det(\Sigma_t)(1 + w_t(\mathbf{y}_t)^2).$$

Telescoping:

$$\det(\Sigma_T) = \det(\Sigma_1) \prod_{t=1}^{T} (1 + w_t(\mathbf{y}_t)^2)$$

$\square$

*Proof of Lemma E.5.* Let the eigenvalues of $\sum_{t=1}^{T} \mathbf{y}_t \mathbf{y}_t^\top$ be $\sigma_1, \sigma_2, \ldots, \sigma_d$. Note that

$$\sum_{i=1}^{k} \sigma_i = \text{Trace} \sum_{t=1}^{T} \mathbf{y}_t \mathbf{y}_t^\top = \sum_{t=1}^{T} \|\mathbf{y}_t\|^2 \leq T.$$

Thus, we have

$$
\begin{aligned}
\log \det \frac{\Sigma_T}{\Sigma_1} &= \log \det \left( 1 + \frac{1}{\lambda} \sum_{t=1}^{T} \mathbf{y}_t \mathbf{y}_t^\top \right) \\
&= \log \left( \prod_{i=1}^{d} \left( 1 + \frac{\sigma_i}{\lambda} \right) \right) \\
&= d \log \left( \prod_{i=1}^{d} \left( 1 + \frac{1}{\lambda} \right) \right)^{1/d} \\
&\leq d \log \left( \frac{1}{d} \sum_{i=1}^{d} (1 + \sigma_i/\lambda) \right) \\
&\leq d \log \left( 1 + \frac{T}{d\lambda} \right),
\end{aligned}
$$

which finishes the proof of the lemma. $\square$

### E.3. Proof of Theorem 5.1

*Proof of Theorem 5.1.* Let $\mathbf{o}$ be the optimal sequence and $\mathbf{g}$ be the resulting sequence from greedy decoding. Suppose not, *i.e.,* $\mathbf{o} \neq \mathbf{g}$ for sake of the contradiction, and let $\mathbf{g}$ be the resulting sequence by the greedy algorithm. Similar to the length equalization process in the proof of Theorem 3.9, we can extend $\mathbf{g}$ and $\mathbf{o}$ to have the same maximal length $L$. Now, it suffices to prove that for any $l \in [L]$, we have $u(\mathbf{o}^{(1:l)}) - u(\mathbf{g}^{(1:l)}) \leq \varepsilon$. We abuse $\text{REG}^{(l)}$ to denote $u(\mathbf{o}^{(1:l)}) - u(\mathbf{g}^{(1:l)})$. Following the arguments in the proof of Theorem 3.9, we can obtain $\text{REG}^{(l)} \leq |\text{REG}^{(l-1)}|$. Thus, we eventually have $\text{REG}^{(l)} \leq \left| \text{REG}^{(i)} \right|$ where $i$ is the first level that $\mathbf{g}$ deviates from $\mathbf{o}$. Since $\text{REG}^{(i)}$ is at most the SLD $\varepsilon$, we finish the proof.

$\square$

### E.4. Proof of Proposition 4.1

*Proof of Proposition 4.1.* It is straightforward to see that $m := |\mathcal{V}^*| = \Omega(n^L)$ since it has (roughly) at least $(n-1)^{L-1}$ sequences that ends with EOS and having depth $L$. Consider an instance of the stochastic $m$-armed bandit problem where each arm $i$ is associated with a distribution $F_i$. Let us (arbitrarily) index each subsequence in $\mathcal{V}_L^*$ by $1, 2, \ldots, m$ and let $i(\mathbf{y})$

be index of $\mathbf{y}$. We construct an instance of TMAB such that each random reward $r_t(\mathbf{y})$ is an independent sample from $F_{i(\mathbf{y})}$. That is, whenever the DM submits a complete sequence $\mathbf{y} \in \mathcal{V}^*$, its random reward is sampled from $F_{i(\mathbf{y})}$ independently. Then, we have that $u(\mathbf{y}) = \int x dF_{i(\mathbf{y})}(x)$. We further assume that if we add more EOS to a complete sequence $\mathbf{y}$, the utility does not change, *i.e.,* have the same distribution. One can easily check that the utillity function satisfies Assumption 2.1 and structurizes $F_i$ so that it satisfies Assumption 2.2 as well.[29] Then, in order to learn a reward for $\mathbf{y}$, any algorithm needs to run a bandit algorithm treating each $\mathbf{y}$ as a single arm. The standard information theoretic argument on the regret lower bound for stochastic bandit problem yields the desired regret lower bound.

$\square$

### E.5. Appendix for TMAB

We require the following variant of DDMC for TMAB.

**Assumption E.6** (DDMC'). *A sequence function $u(\cdot)$ has* diminishing distance with more commons *(DDMC') if*

$$| u(x_t, \mathbf{y} : \tau : \texttt{EOS}) - u(x_t, \mathbf{z} : \tau : \texttt{EOS}) | \leq | u(x_t, \mathbf{y} : \texttt{EOS}) - u(x_t, \mathbf{z} : \texttt{EOS}) |,$$

*for any two different finite sequences $\mathbf{y}, \mathbf{z}$ with the same length $|\mathbf{y}| = |\mathbf{z}|$ and any token $\tau \in \mathcal{V}$.*

*Proof of Theorem 4.2.* Overall, we will prove that at any depth $k$ during the execution algorithm, it will select the optimal subsequent token for depth $k + 1$ with high probability. Let us begin with depth 1 for reader's convenience. Note that $\bar{r}(\mathbf{y} : \tau : \texttt{EOS})$ denotes the average of the cumulative reward obtained from exploring $\mathbf{y} : \tau : \texttt{EOS}$ for $N$ rounds during the execution of the algorithm. For any $\mathbf{y} \in \mathcal{V}^*$, define the clean event as follows:

$$\texttt{Clean}_{\mathbf{y}} = \{|\bar{r}(\mathbf{y}) - u(\mathbf{y})| \leq \sqrt{2 \log T / N}\}.$$

By Hoeffding inequality, we have

$$\Pr\left[\texttt{Clean}_{\mathbf{y}}\right] \geq 1 - 2/T^4,$$

for any $\mathbf{y} \in \mathcal{V}^*$. Suppose our algorithm chooses token sequence $\mathbf{a}$ whereas the optimal algorithm chooses $\mathbf{o}$. If $\mathbf{o} = \texttt{EOS}$, then the problem boils down to a single-level scenario and the standard analysis of ETC algorithm carries over. Assume $\mathbf{o} \neq \texttt{EOS}$, *i.e.,* $\mathbf{o}$ considers at least one non-EOS token.

Due to our algorithm's choice, $\bar{r}(\mathbf{a}^{(1)} : \texttt{EOS}) \geq \bar{r}(\mathbf{o}^{(1)} : \texttt{EOS})$. Define

$$\texttt{Clean}^{(1)} = \cap_{\tau \in \mathcal{V}} \texttt{Clean}_{\emptyset : \tau : \texttt{EOS}},$$

*i.e.,* the clean event over every token at depth 1. Taking union bound over $\mathbf{y} = \emptyset$ and $\tau \in \mathcal{V}$, we have

$$\Pr\left[\texttt{Clean}^{(1)}\right] \geq 1 - 2/T^3.$$

Under this event, we know that

$$
\begin{aligned}
u(\mathbf{a}^{(1)} : \texttt{EOS}) + \sqrt{2 \log T / N} &\geq \bar{r}(\mathbf{a}^{(1)} : \texttt{EOS}) \\
&\geq \bar{r}(\mathbf{o}^{(1)} : \texttt{EOS}) \\
&\geq u(\mathbf{o}^{(1)} : \texttt{EOS}) - \sqrt{c \log T / N}.
\end{aligned}
$$

Thus, with probability $1 - 2/T^3$, we obtain $u(\mathbf{o}^{(1)}) - u(\mathbf{a}^{(1)}) \leq 2c \log T / N$. Define $d := \sqrt{2 \log T / N}$.

Now we will generalize this argument to prove that we have $u(\mathbf{o}^{(1:l)} : \texttt{EOS}) - u(\mathbf{a}^{(1:l)} : \texttt{EOS}) \leq d$ under an appropriate clean event occurring with high probability. Note that this is sufficient since $u(\mathbf{o}^{(1:l)} : \texttt{EOS}) = u(\mathbf{o}^{(1:l)})$ once $\mathbf{o}^{(1:l)}$ reaches EOS, and analogously for $\mathbf{a}$.

Subtly, we need to handle the case where $\mathbf{o}$ and $\mathbf{a}$ have different lengths. Similar to previous analysis, we equalize the length by adding more EOS tokens so that both $\mathbf{o}$ and $\mathbf{a}$ has length $L$. Then, we can safely deal with the extended sequences with

---

[29]We omit the details as it is a cumbersome argument.

the same length. We prove by induction on the length $l$ of the subsequence of $\mathbf{o}$ and $\mathbf{a}$ by comparing $\mathbf{a}^{(1:l)}$ with $\mathbf{o}^{(1:l)}$. For every $i \in [L]$ and $i \geq 2$, we recursively define the following events:

$$\texttt{Clean}^{(1:i)} = \left(\cap_{j \in [i-1]}\texttt{Clean}^{(1:i-1)}\right) \cap \left(\cap_{\tau \in \mathcal{V}}(\texttt{Clean}_{\mathbf{a}^{(1:i-1)}:\tau:\text{EOS}})\right) \cap \texttt{Clean}_{\mathbf{o}^{(1:i-1)}:\text{EOS}},$$

and define $\texttt{Clean}^{(1)}$. By abusing $\texttt{Clean}^{(1:0)} = \emptyset$ and $\mathbf{o}^{(1:0)} = \mathbf{a}^{(1:0)} = \emptyset$, note that we have $\texttt{Clean}^{(1:1)} = \texttt{Clean}^{(1)}$.

Let $k$ be the first level that $\mathbf{a}$ deviates from $\mathbf{o}$. Now, assume for the sake of induction that $u(\mathbf{o}^{(1:i)} : \text{EOS}) - u(\mathbf{a}^{(1:i)} : \text{EOS}) \leq 2d + \left|\text{REG}_t^{(k)}\right|$ holds under the clean event $\texttt{Clean}^{(1:i)}$. Now we will prove that under $\texttt{Clean}^{(1:i+1)}$, we have $u(\mathbf{o}^{(1:i+1)} : \text{EOS}) - u(\mathbf{a}^{(1:i+1)} : \text{EOS}) \leq 2(i+1)d + \left|\text{REG}_t^{(k)}\right|$.

Similar to the analysis of EOFUL, we consider a fictitious extension of $\mathbf{f} = \mathbf{a}^{(1:i)} : \mathbf{o}^{(i+1)}$.

Due to the construction of clean event and our algorithm's optimistic choice, we have

$$u(\mathbf{a}^{(1:i+1)} : \text{EOS}) + \sqrt{c \log T/N} \geq \bar{r}(\mathbf{a}^{(1:i+1)} : \text{EOS})$$
$$\geq \bar{r}(\mathbf{f} : \text{EOS})$$
$$\geq u(\mathbf{f} : \text{EOS}) - \sqrt{c \log T/N}.$$

If $u(\mathbf{o}^{(1:i+1)} : \text{EOS}) - u(\mathbf{f} : \text{EOS}) \leq 0$, then the induction follows. Otherwise, by DDMC', we have

$$u(\mathbf{o}^{(1:i+1)} : \text{EOS}) - u(\mathbf{f} : \text{EOS}) \leq \left|u(\mathbf{o}^{(1:i)} : \text{EOS}) - u(\mathbf{a}^{(1:i)} : \text{EOS})\right|$$
$$\leq 2id + \text{REG}_t^{(i)},$$

and combining the inequalities we have $\text{REG}_t^{(i+1)} \leq 2(i+1)d + \left|\text{REG}_t^{(k)}\right|$.

Thus, by the induction principle, under $\texttt{Clean}^{(1:L)}$, we have $u(\tilde{\mathbf{o}}^{(1:L)}) - u(\tilde{\mathbf{a}}^{(1:L)}) \leq 2Ld + O(1/\sqrt{T})$ as $\text{REG}_t^{(k)} = O(1/\sqrt{T})$ due to the SSLD assumption. For ease of exposition, we will simply use the upper bound $\text{REG} \leq 2LD$ as $O(1/\sqrt{T})$ contributes at most $\sqrt{T}$ regret by naively multiplying by $T$.

Recall the definition of $\texttt{Clean}^{(1:L)}$:

$$\texttt{Clean}^{1:L} = \left(\cap_{j \in [L-1]}\texttt{Clean}^{(1:L-1)}\right) \cap \left(\cap_{\tau \in \mathcal{V}}(\texttt{Clean}_{\mathbf{a}^{(1:L-1)}:\tau:\text{EOS}})\right) \cap \texttt{Clean}_{\mathbf{o}^{1:L-1}:\text{EOS}}$$
$$= \prod_{t=1}^{L} \left(\cap_{\tau \in \mathcal{V}}(\texttt{Clean}_{\mathbf{a}^{(1:t-1)}:\tau:\text{EOS}}) \cap \texttt{Clean}_{\mathbf{o}^{1:t-1}:\text{EOS}}\right),$$

where we abuse $\mathbf{a}^{(1:0)}$ and $\mathbf{o}^{1:0}$ to denote $\emptyset$. By the independence of the reward tapes and union bound, we have

$$\Pr\left[\cap_{\tau \in \mathcal{V}}(\texttt{Clean}_{\mathbf{a}^{(1:t-1)}:\tau:\text{EOS}})\right] \geq 1 - \frac{2}{T^4} \cdot n.$$

Thus, again by union bound, we have

$$\Pr\left[\texttt{Clean}^{(1:L)}\right] \geq 1 - \frac{2(n+1)L}{T^4} \geq 1 - \frac{4L}{T^4}$$

Now the entire regret can be written as:

$$\text{REG} = \text{REG}_{\text{exploration}} + \text{REG}_{\text{exploitation}}$$
$$\leq n \cdot L \cdot N \cdot 1 + \text{REG}_{\text{exploitation}}$$
$$\leq nLN + 2LTd \cdot \Pr\left[\texttt{Clean}^{(1:L)}\right] + T \cdot \Pr\left[\neg\texttt{Clean}^{1:L}\right]$$
$$\leq nLN + 2LT\sqrt{\frac{2\log T}{N}} + T\frac{4nL}{T^4}.$$

Assuming that $n, L \leq T$, plugging $N = T^{2/3}(\log T)^{1/3}$ yields the regret of $\text{REG} = O(nLT^{2/3}(\log T)^{1/3})$.[30] $\qquad\square$

---

[30] If $n$ or $L$ is large compared with $T$, one can simply increase the exploration parameter in the algorithm and neutralize such dependency.

