# OpenReview forum: "Tokenized Bandit for LLM Decoding and Alignment"
_ICML.cc/2025/Conference — ICML 2025 poster_

### Official Review · Reviewer_dJHj · 2025-02-14

**Overall Recommendation:** 2

**Summary:**

The paper "Tokenized Bandit for LLM Decoding and Alignment" introduces a novel Tokenized Bandit (TB) framework to address LLM decoding and alignment challenges. It models LLM decoding as a sequential decision-making problem, using multi-armed bandit (MAB) and linear bandit (LB) techniques to optimize token selection.

**Claims And Evidence:**

The DDMC assumption is not universally validated. It is tested on limited datasets, and it's unclear whether it holds for other LLM applications (e.g., coding, mathematical reasoning).

**Essential References Not Discussed:**

N/A

**Experimental Designs Or Analyses:**

1. While the paper claims that bandit-based decoding is superior, it does not compare against reinforcement learning approaches, which are common in LLM alignment. For example, Best-of-N Sampling and Direct Preference Optimization (DPO) are stronger baselines for alignment than standard greedy decoding.

2. I worry about the practical applicability of the approach: Tokenized Bandit selects tokens step by step but assigns rewards at the sequence level, making it hard to attribute rewards to individual tokens. Moreover, Bandit algorithms assume independent token selection, while **LLM generation has strong sequential dependencies**, raising doubts about their effectiveness in optimizing decoding. In practice, greedy decoding is rarely used due to repetitive outputs, possibly caused by the independence assumption in token selection.

**Methods And Evaluation Criteria:**

Expanding evaluations to more datasets, and efficiency benchmarks  (e.g., PPO, Best-of-N decoding) would significantly strengthen the paper’s claims.

**Other Comments Or Suggestions:**

1. I could not find the explicit formulation of the embedding function e(x_t, y). Does it lack an explicit expression? Or can you explain how to evaluate whether the token sequence y is suitable to query x_t, i.e., where is the reward r come from?

2. Regarding the definition of the utility function, could the authors provide some intuitive examples to justify its formulation? The current definition seems somewhat unconventional.

3. Concerning the experimental results, in the second figure, the convergence result appears to be around 20, which does not seem very satisfactory. I suspect that a significant factor contributing to this issue might be the formulation or definition of the utility function. Could the authors elaborate on whether any alternative formulations were considered?

4. Minor error: (1) In algorithm 2, there is a mistake in the selection of \tau^{\star}. (2)For y^{(i)}, it is the i-th token in sequence \bm{y}, but you use boldface on it, is it a typo?

**Other Strengths And Weaknesses:**

Other Strengths:
1. The paper also introduces the Diminishing Distance with More Commons (DDMC) hypothesis, which states that if two sequences share the same suffix, their reward difference decreases. This assumption significantly reduces computational complexity and justifies the effectiveness of greedy decoding.
2. The paper theoretically proves that greedy decoding can be globally optimal under the DDMC hypothesis.

Other Weaknesses:
1. The paper assumes that the reward function 𝑢ₜ(𝑥ₜ, 𝑦) follows a linear structure:  $𝑢ₜ(𝑥ₜ, 𝑦) = ⟨𝜃, 𝑒(𝑥ₜ, 𝑦)⟩$. However, in real-world LLM tasks, user preferences 𝑓(𝑥ₜ, 𝑦) may exhibit highly nonlinear behavior.
2. In the Tokenized Multi-Armed Bandit (TMAB) setting, the reward can only be observed after generating the entire sequence, making it difficult to directly determine the contribution of individual tokens. To address this, the authors adopt the Explore-Then-Commit (ETC) approach, estimating the utility of each token through multiple full-sequence samples. However, I am concerned that the number of required full sequences might be excessively large, potentially leading to a high computational cost.

**Questions For Authors:**

1. In the exploration phase of GreedyETC (Greedy Exploration-Then-Commit), how is the reward assigned to individual tokens when a full sequence must be generated before obtaining a reward? Since the reward is only observed at the sequence level, how do you estimate the contribution of each token during exploration?

2. The paper formulates LLM decoding as a Tokenized Bandit problem, where each token is selected sequentially, and the final reward is only observed after a complete sequence is generated. Given this delayed reward structure, how does the proposed bandit-based approach achieve global sequence-level optimization? Would a Combinatorial Bandit or Reinforcement Learning (RL) approach be more appropriate for handling dependencies between tokens?

3. It is unclear how a bandit approach, which is typically state-independent, can effectively guide token selection when the reward is assigned only after the entire token sequence $y_t$  is generated. Since token selection is not a set-based decision but rather a structured ordered sequence, different sequences lead to vastly different rewards, and the contribution of each token is not independently determinable. How does the bandit framework, which does not model dependencies between token choices, provide meaningful guidance in this setting?

**Relation To Broader Scientific Literature:**

The paper should expand the discussion on prior search-based decoding, RLHF baselines, and uncertainty-aware text generation to strengthen positioning in the broader literature.

**Theoretical Claims:**

The regret bound relies on DDMC, but the paper does not rigorously prove whether DDMC always holds across different tasks.
And the proof assumes perfect token embeddings $e(x_t,y)$, but in reality, embeddings could introduce variance that affects regret analysis.

---

> ### Author Rebuttal · Authors · 2025-03-30
>
> We truly appreciate your insightful comments.
> We will first make a general remark on our main contribution, answer major concerns and then remaining questions.
>
> # General remark on main contribution / assumption
>
> We kindly refer to our **response (General remark on main contribution) to Reviewer 3dxc**.
>
> # Major concerns
>
> ## 1. Individual token reward
> We note that individual tokens **do not** have any rewards, but only a token sequence does.
> Our algorithm smartly learns an efficient path to the optimal sequence only by observing seqeunce-wise rewards, thanks to the  novel DDMC assumption.
>
> ## 2. Sequential dependency
>
> Our settings intrinsically **considers sequential dependency** given what has been chosen previously, since these histories are encoded in the sequence function’s value and also final embedding vectors.
> Indeed, our offline benchmark selects a sequence with the largest utility that does account for the sequential dependency of the LLM, and the regret is computed with respect to this.
> Thus, our goal is to learn efficient ‘sequence’ not individual token, and the previous decisions affect onwards.
>
> We also remark that greedy decoding often provides repetitive outputs, especially for small language models few years ago, but this phenomenon is mitigated in the most recent LLMs (Song et al. 2024).
> Practically, one can replace greedy decoding with other decoding schemes in EOFUL algorithm based on its empirical performance, which will be a great direction for future work.
>
> ## 3. Comparison with other benchmarks - PPO, DPO, Best of N
> For PPO and DPO, we refer to **our response (Comparison to ARGS / PPO in RLHF) to Reviewer 3dxc**.
>
> For the Best-of-N alignment, one could consider Best-of-N algorithm that knows the latent reward function in hindsight in our regret definition as an offline benchmark (note that this is **not an online learning**).
> However, this will be weaker benchmark than offline greedy decoding under DDMC assumption.
> Practically, one can replace the greedy decoding with best-of-n (or other decoding schemes) in EOFUL algorithm.
>
> # Further comments
>
> ## 1. Perfect embeddings
>
> Once the regret benchmark and a learning algorithm share the same embedding vector (even perturbed), all our theoretical results would naturally follow.
> Also, standard LLMs operate based on the embedding vectors and corresponding logit probabilities, so the algorithm can naturally access embedding vectors, which is how we obtain embeddings in our evaluations.
>
> ## 2. State-dependency
> Our contextual bandit framework captures state-dependency, and we hope our response (Sequential dependency) resolves your question.
>
> ## 3. Comments on DDMC with more datasets
> We kindly refer to our response (More justification on DDMC) to Reviewer 3dxc.
>
> ## 4. Linear realizability assumption
> We kindly refer to our response (Comments on linear realizability) to Reviewer amHn.
>
> ## 5. ETC approach in TMAB
> Our study on TMAB is to provide more comprehensive theoretical findings for ‘tokenized’ variants of the bandit problems, whereas we believe our TLB setting is more practical for LLM applications.
>
> That said, the fact that greedy ETC achieves the regret sublinear in $T$ and linear in the length of the sequence in TMAB implies that the number of samples required to learn efficient sequence is not significant.
> We note that a naive algorithm requires exploring every possible sequence inducing regrets exponential in the length of the sequence.
>
> ## 6. Using combinatorial bandit or RL
> As our objective function is a sequence function not a set function, standard combinatorial bandits can't be directly applied.
> Further, one may generalize TLB to Markov decision process as does in RL, but this would be waymore complicated to analyze and it is not certain if it would exhibit Markovian property.
>
> ## 7. Explicit formulation of the embedding function
> We hope our response (Perfect embedding) resolves your question.
> Also, a noisy reward is observed after the LLM submits the entire outputs y, similar to the standard bandit framework, e.g., see L#146 (right column) on page 3.
>
> ## 8. Utility function
> We hope our response (Linear realizability assumption) resolves your question.
> Also, the weighted average of original LLM’s probability and another external reward module’s score is often considered for LLM alignment, e.g., see Khanov et al., 2024.
>
> ## 9. Convergence on the experiment results
> Our interest is not on the convergence, but whether it exhibits a decreasing trend with respect to more common tokens as the definition of DDMC.
>
> ## 10. Minor error: (1) In algorithm 2, there is a mistake in the selection…
> Thanks for catching - will edit it. Also yes, $y^{(i)}$ is $i$-th token (single token) - we abuse boldface to make it consistent with $y^{(i:j)}$ which is a sequence.
>
>
> # References
>
> * Song et al. 2024, The Good, The Bad, and The Greedy: Evaluation of LLMs Should Not Ignore Non-Determinism
> * Khanov et al. 2024, Alignment as Reward-Guided Search

---

> > ### Comment · Reviewer_dJHj · 2025-04-02
> >
> > I’m not sure if I’ve missed something here. Due to space constraints, the author’s responses can’t be overly detailed, but I’d like the author to provide a clearer explanation: how do they address the issue of strong sequential dependencies? namely, how is the reward or feedback for the intermediate token selection process determined, since a reward is only obtained after generating an entire sentence? This problem is somewhat similar to the distinction in LLM reasoning between outcome and process reward models. Defining rewards for intermediate steps in a process reward model is notoriously tricky—here, that’s equivalent to assigning a reward for each token selection. Notably, this paper isn't relying on training; we’re just using a lightweight bandit approach. I still don’t fully grasp how the author resolves this challenge.

---

> > > ### Author Response · Authors · 2025-04-03
> > >
> > > We sincerely appreciate the reviewer for detailed questions.
> > > We will give an intuition behind how our algorithm and assumptions enable efficient learning without intermediate feedback, focusing on the TLB setting.
> > > Note that the arguments for TMAB follow similar intuition, though details are different.
> > >
> > >
> > > # Intuition off the top of one's head
> > > In short, efficient learning is possible via (0) structural assumptions on DDMC and linearly realizable reward, (i) fact that offline greedy decoding is optimal under DDMC, (ii) we make an estimator $\tilde{\theta}_t$ for $\theta$ each round given feedbacks and use it for the decoding, (iii) the combination of LinUCB style of algorithm plus greedy decoding incurs only a small error for each token-wise decision compared to the offline greedy decision given that the estimator $\tilde{\theta}$ is not very far from $\theta$, and (iv) aggregating all these conclude efficient regrets bounds.
> > >
> > > Notably, structural assumptions imposed on the sequence reward function by **DDMC assumption** and **linear utility function** with respect to embedding vectors enable us to efficiently decode tokens to find efficient sequence.
> > > One may interpret the intermediate reward as the expected utility of the subsequence given estimators of the latent parameters.
> > > Alternatively, rather than considering arbitrary reward function of the LLM, we show that if the utility (reward) satisfies reasonable structural assumptions, one can efficiently learn it in decoding time with online feedback.
> > > Thus, our results may be also of interest to LLM reward modeling in handling the trade-off between deploying simple model versus efficiency.
> > >
> > > EDIT: To clarify a bit more, in general situation, credit assignment is notoriously hard as the reviewer pointed out. But in online learning scenario, certain amount of errors are inevitable as we need to learn the latent parameters on the fly, and the objective is to have cumulative errors negligible w.r.t. number of samples (ideally sublinear in $T$) and this becomes possible as presented above.
> > >
> > > # Overall arguments
> > >
> > >
> > > Our overall arguments proceed as follows:
> > > 1. We prove that offline greedy (that knows the $\theta$) is optimal under DDMC.
> > > 2. Thus, our algorithm's objective boils down to mimic the offline greedy algorithm's behavior sample-efficiently.
> > > 3. Mainly, we combine the LinUCB style of algorithm with greedy decoding scheme that constructs an estimator $\tilde{\theta}_t$ and confidence ball $C_t$ using available history, and use it to decode at round $t$.
> > > 4. We prove that for every token decision above, the token-wise regret is sufficiently small as a function of the radius of $C_t$ and error of $\tilde{\theta}_t$.
> > > 5. Aggregating these error terms carefully with appropriate parameters, the regret can bounded by $L\sqrt{T \ln T}$.
> > >
> > > We note that we slightly revise the arguments for sake of understanding, and is a bit different from the actual steps.
> > > We refer to (L#289, page 6) below Theorem 3.8 and proofs for more details.
> > >
> > >
> > > # Detailed explanation for each argument
> > >
> > > 1. Offline greedy is optimal under DDMC
> > >
> > > First, we show that if we know the latent parameter $\theta$, greedy decoding is optimal under DDMC.
> > > Here's a rough proof sketch.
> > > Suppose not and say $o$ is an optimal sequence and $g$ is the sequence created by greedy decoding such that $u(o) > u(g)$.
> > > Suppose $o$'s length is $L$.
> > > If $L = 1$, then contradiction since $g$ makes myopically optimal decision.
> > > Then, for induction argument we assume $g$ is optimal for $L-1$, and validate $L$.
> > > This is done by comparing $o^{(1:L)}$ with a sequence $f$ that appends $o^{(L)}$ upon $g^{(1:L-1)}$ and utilizing DDMC assumption as well as the greedy decoding's nature.
> > >
> > > 2. Objective becomes mimicking greedy
> > >
> > > Thus, if algorithms can make exact greedy choices for each token, then regret is zero.
> > > On the other hand, since we don't know $\theta$, we need to learn it efficiently.
> > >
> > > 3. LinUCB + greedy decoding
> > >
> > > - Once we have round $t-1$ feedback, we construct an estimate $\tilde{\theta}_t$ and confidence ball $C_t$.
> > > - Then, for the next round $t$'s first token, we select the token that maximizes LinUCB index.
> > > - This procedure is repeated for every token by appending the new token to the previously selected tokens and compute LinUCB index given corresponding embeddings.
> > > - After it reaches EOS token, we submit and observe the feedback, and repeats.
> > >
> > > 4. Token-wise regret
> > >
> > > Finally, total regret is decomposed by the summation of regrets over each round/token.
> > > Fix a round $t$. For $l$-th token selection, by DDMC assumption, we relate the error made at $l$-th token selection to the error made at $l-1$-th token selection. Telescoping, the round $t$ regret can be written as a function of the error made at the first token selection, which can be written as the error rates of $\tilde{\theta}$ and the radius of $C_t$.
> > >
> > > 5. Aggregating round-wise regrets
> > >
> > > Using algebraic arguments, we conclude that summing up these regrets overall induce sublinear regret.

---

### Official Review · Reviewer_amHn · 2025-02-21

**Overall Recommendation:** 3

**Summary:**

The paper introduces two new bandit variants, the tokenized linear bandit (TLB) and tokenized multi-armed bandit (TMAB), which involve sequentially constructing a sequence of tokens to optimize a (random) utility function of a user, given a query. They introduce the DDMC assumption on token sequences and construct learning algorithms for both TLB and TMAB for which they prove regret bounds. They show that LLM alignment may be viewed as an application of their TLB method for specific functional forms of utility misalignment between the model and the user. They also aim to empirically validate the DDMC assumption in the context of LLMs and connect it to the empirically observed high performance of greedy decoding in LLMs.

## update after rebuttal
Following the author response, I am *provisionally* increasing my score to a 3, subject to discussion with the other reviewers.

**Claims And Evidence:**

I have not been able to evaluate all the proofs, but claims seem generally well supported, with a few exceptions.

My first worry is regarding the suggestion that this method may be useful for alignment. It seems that the TLB problems only applies insofar as the linear realizability assumption holds, as well as the particular functional form of misalignment suggested in section 5.1. The paper contains no empirical verification of this in typical alignment settings.

My second worry relates to the DDMC assumption. The validity of the assumption is evaluated assuming the utility is parameterized to fit the form of distance functions d1 or d2 (sec 5.3), and in those cases it broadly (though not always, inspecting figure 1) seems to hold on Llama-3-8B-Instruct for the two datasets tested. However, this is not evaluated on more free form utility functions, such as those provided by human subjects or even reward models. In particular, it seems this assumption cannot hold in general (as shown in figure 1), and in particular it seems to omit some rather standard use cases of LLMs.

For instance, take the following token sequences:

q = "Suggest a cafe I should visit in Vancouver.",
y = "Visit Cafe Alice",
z = "Visit Cafe Bob",
\tau = "by"

y and z could plausibly be equal token length (this depends on the tokenizer, which is not part of the assumption). If 'Cafe Bobby' exists and 'Cafe Aliceby' does not exist, then utility z + tau is presumably larger than y + tau, even thought utility of y and z were presumably equal. Thus, the gap has increased.

In the case that u = p, the DDMC assumption implies that the difference in probability under the LLM of y and z is smaller than that of y + tau and z + tau, which seems unreasonable if 'Cafe Bobby' exist and 'Cafe Aliceby' does not, but 'Cafe Bob' and 'Cafe Alice' both exist. This suggests that p does not fit theorem 5.1, so the claimed connection to the success of greedy LLM decoding seems somewhat unreliable.

Note that this is dependent on the state of the world, which is not part of the assumption statement for obvious reasons, but without it the statement makes potentially unsupported assumptions about the world. The general issue is that tokens mean very different things in different contexts, so sharing more tokens does not guarantee that utilities are closer in many cases.

**Essential References Not Discussed:**

Nothing particular that I am aware of. There is a wide literature on decoding time alignment that was not discussed (e.g., steering methods such as Panickssery et al. 2023, "Steering Llama 2 via Contrastive Activation Addition" and Todd et al. 2023, "Function Vectors in Large Language Models"), but it does not relate to the methods proposed in this work too directly.

**Experimental Designs Or Analyses:**

See previous fields.

**Methods And Evaluation Criteria:**

Testing the DDMC assumption with Llama-3-8B-Instruct on the TruthfulQA and HH-RLHF datasets is sensible. However, as mentioned above, the utility here is assumed to be parameterized in two very specific ways, and it would be instructive to see whether DDMC holds empirically with more free form utility functions, such as those provided by human subjects or even reward models.

**Other Comments Or Suggestions:**

Typo in proposition 3.3: "assumptinon"

**Other Strengths And Weaknesses:**

Strengths:
- The paper is generally well written and structured.
- The paper contains many notes pointing to related work and exact implications of assumptions and statements. The authors seem very aware of the mathematical literature, although this is hard for me to reliably evaluate.

Weaknesses:
- Primarily the degree to which the assumptions (DDMC and to a lesser extent linear realizability of utility functions) hold in practice and the absence of (compelling) experimental evidence for these assumptions and the application of the work to alignment.

**Questions For Authors:**

1. Under what conditions can we expect DDMC to hold, and is this realistic for typical LLM use cases?

**Relation To Broader Scientific Literature:**

The paper proposes two new discrete bandit problems with applications to LLM decoding and alignment. I am unaware of earlier works on this intersection, but I am not too familiar with that literature. The paper aims to provide evidence towards a mechanism for the success of greedy decoding, which is a topic of current interest given the ubiquity of LLMs. For the same reason, results on decoding time alignment are relevant.

**Theoretical Claims:**

I have not been able to evaluate the proofs.

---

> ### Author Rebuttal · Authors · 2025-03-30
>
> We truly appreciate your detailed feedback and insightful comments.
> We will first make a general remark on our main contribution, and then answer the reviewer’s comments/questions one by one.
>
> # General remark on main contribution
>
> We kindly ask the reviewer to see **our response (General remark on main contribution) to Reviewer 3dxc**.
>
> # Comments on DDMC
> We hope our general remark above partly resolves the reviewer’s concern regarding the necessity of the assumption.
>
> As the reviewer pointed out, we don’t believe that DDMC would universally hold for every real-world scenario.
> However, we empirically find an overall tendency to do so, i.e., having a decreasing utility difference (or decreasing distance between embedding vectors) as we append more common tokens.
>
> Our DDMC assumption, or at least a relaxed version of it, is quite reasonable intuitively. For instance, in extreme cases, if we append a lot of common tokens in each of two sequences of the same length, one could expect that the user will realize a similar experience of reading them. Further, as we discussed in L#234, page 5, this shares similar intuition from the widely adopted submodularity on set functions.
>
> Also, we conducted a few **more experiments** to cement our assumption’s effectiveness.
> It can be found in https://anonymous.4open.science/r/temp-03AC.
> In new experiments, we first tested DDMC assumption on two more datasets: AdvBench (standard jailbreak benchmark) and just-eval-instruct that contains many prompts of various tasks. Further, we verified DDMC assumptions beyond our linear utility function. We considered three more functions: L1/3/4 distance. In all these scenarios, we observe a tendency that appending more common tokens decreases the utility gaps. We will certainly add these experiments in the revision.
>
>
> # Under what conditions can we expect DDMC to hold...
>
> Thanks for a great question.
> Although it’s difficult to classify certain characteristics of tasks that DDMC would more strongly hold, through our empirical verifications (including **newly added ones**), we have identified that in several commonly-used LLM alignment datasets, DDMC assumption seems to hold, so we believe this assumption is reasonably realistic (though certainly not universal).
>
> # Comments on linear realizability
>
> We hope our general remark above partly resolves the reviewer’s concern regarding the necessity of the assumption.
>
> As we noted in our manuscript (paragraph below Assumption 3.2), several papers in the literature for LLM alignment often assume linear realizability for theoretical sake, e.g., we refer to Cen et al. 2024 and Yang et al. 2024 that exactly assume the linear realizability for theoretical purpose in LLMs
>
> From practical perspectives, **several** recent empirical studies provide evidence that linear realizability is reasonable in many scenarios.
> First, Zou et al. 2025 confirmed that concepts like truthfulness or ethics could be extracted via linear transformation over LLM’s representation (embedding) with fixed weights, which is also validated through extensive experiments.
> Wang et al. 2024 tackled decoding-time alignment solely by using linear transformation over the LLM’s representation with carefully chosen weights to handle the harmlessness alignment.
> Kong et al. 2024 considered a control theoretical perspective for LLM alignment by adding a linear perturbation on original logit vector.
>
> Another relevant literature is the recently studied ‘linear representation hypothesis’, which gives a mechanistic interpretation on how LLMs embed concepts in representation space, as noted in our footnote 14.
> In particular, the hypothesis itself states (as hinted by its name) that concepts / semantics (e.g., politic ideology, geography, and temporal knowledge) are embedded in a linear manner given some proper weights: we refer to Kim et al. 2025, Jiang et al. 2024, Park et al. 2025, Gurnee and Tegmark 2024 for more details.
>
> Finally, we believe extending the linear function to nonlinear function in a similar vein to the extension of linear contextual bandit to nonlinear contextual bandit via kernelization by Valko et al. 2013 would be a great direction for future works.
>
>
>
>
>
> # References
>
> * Cen et al. 2024, Unified Approach to Online and Offline RLHF
> * Yang et al. 2024, Asymptotics of Language Model Alignment.
> * Zhou et al. 2025, Representation Engineering: A Top-Down Approach to AI Transparency
> * Wang et al. 2024, InferAligner: Inference-Time Alignment for Harmlessness through Cross-Model Guidance
> * Kong et al. 2024, Aligning Large Language Models with Representation Editing: A Control Perspective
> * Kim et al. 2025, Linear Representations of Political Perspective Emerge in Large Language Models
> * Jiang et al. 2024, On the Origins of Linear Representations in Large Language Models
> * Park et al. 2025, The Geometry of Categorical and Hierarchical Concepts in Large Language Models
> * Gurnee and Tegmark 2024, Language Models Represent Space and Time

---

### Official Review · Reviewer_9ZG7 · 2025-03-13

**Overall Recommendation:** 3

**Summary:**

The paper introduces the Tokenized Linear Bandit (TLB) and Tokenized Multi-Armed Bandit (TMAB), which are variants of the classical linear and stochastic multi-armed bandit problems, inspired by the decoding and alignment processes in large language models (LLMs). In these problems, a user submits a query (context) in each round, and the decision maker (DM) sequentially selects tokens irrevocably from a predefined token set. Once the sequence is completed, the DM receives a random utility from the user, whose expected value is determined by a sequence function that maps the chosen token sequence and the query to a non-negative real number.

**Claims And Evidence:**

This paper make interesting claims which are suppoted by sufficient and strong evidences.

**Essential References Not Discussed:**

Essential references have been properly discussed.

**Experimental Designs Or Analyses:**

Experiments are conducted to validate the empirical performance of the proposed methods.

**Methods And Evaluation Criteria:**

The proposed methods and evaluation criteria are well-reasoned. However, numerical results demonstrating their performance in real applications were not provided.

**Other Comments Or Suggestions:**

None

**Other Strengths And Weaknesses:**

None.

**Questions For Authors:**

None

**Relation To Broader Scientific Literature:**

The theoretical framework of tokenized bandits is novel, and the approach used to solve the problem is innovative.

**Theoretical Claims:**

The theoretical claims appear to be correct. Specifically, this work introduces a significant framework that effectively models the LLM decoding and alignment problem. While the problem itself is inherently challenging, it is addressed through an elegant assumption of DDMC, which provides a clever and practical solution.

---

> ### Author Rebuttal · Authors · 2025-03-30
>
> We truly appreciate your detailed feedback and insightful comments, in particular we are glad that the reviewer enjoys our problem and approach.
>
> As the reviewer suggested, we agree that numerical results demonstrating our algorithm’s performance would greatly improve our paper - we appreciate your suggestion.
> As such, we **evaluate our algorithm’s regret** with several benchmarks that we thought to be reasonable in our scenario.
> We refer the reviewer to see: https://anonymous.4open.science/r/temp-03AC, directory /Regrets.
>
> We evaluate algorithms based on **TLB model and LLM alignment scenario** presented in Section 5.1 with $\theta = [1/2,1/2,\ldots, 1/2]$, maximal length of sequence = 30, rounds 5000, and $\gamma = 0.8$.
> We compare our EOFUL along with a benchmark and two other algorithms: (i) theoretical regret upper bound (under-scaled by 0.1 to make the plot more visible), (ii) WrongTheta that uses wrongly estimated $\theta = [-1/2,-1/2,\ldots,-1/2]$ and greedily decoding based on the weighted score, and (iii) Misaligned greedy that only greedily decodes with respect to the LLM’s probability.
> Also, for practicality and computational efficiency, we only consider the top 15 tokens given by LLM in deciding the next token.
> As observed in the figure, we observe that EOFUL effectively achieves sublinear regret.
> Given that the regret upper bound is under-scaled,the actual performance may be much better than that of the theoretical guarantees.

---

> > ### Comment · Reviewer_9ZG7 · 2025-04-07
> >
> > Thank you for the responses to the reviewers. I really enjoyed reading this work and am inclined to accept it. That said, I believe there is still room for improvement, particularly by enriching the experimental section with more general cases (I noticed that similar concerns were also raised by other reviewers). Therefore, I will maintain my current score.

---

### Official Review · Reviewer_3dxc · 2025-03-17

**Overall Recommendation:** 3

**Summary:**

This paper introduces Tokenized Linear Bandit (TLB) and Tokenized Multi-Armed Bandit (TMAB), two variants of bandit algorithms designed for LLM decoding and alignment. These frameworks model LLM decoding as a sequential decision-making process, where a decision-maker selects tokens iteratively to form a complete sequence, receiving a random utility score based on user preferences. The authors establish that learning is impossible without structural assumptions and introduce a key assumption called Diminishing Distance with More Commons (DDMC), which enables efficient learning. They propose EOFUL (Excessive Optimism under the Face of Uncertainty) for TLB and GreedyETC (Greedy Exploration-Then-Commit) for TMAB, achieving sublinear regret bounds. A major theoretical insight is that greedy decoding can be optimal under DDMC, providing a justification for its effectiveness in LLM decoding. Empirical validation on Llama-3-8B with TruthfulQA and HH-RLHF datasets supports the DDMC assumption.

---
### Update after rebuttal
```
As mentioned further below, I am happy with the responses the authors provided. I think there is value in accepting this paper. Hoping there will be consensus among the reviewers that this is indeed the case.
```
---

**Claims And Evidence:**

The paper presents Tokenized Bandit frameworks (TLB and TMAB) for LLM decoding and alignment and supports its claims using rigorous theoretical analysis and empirical validation. Most of the claims are derived with clear mathematical proofs and sublinear regret bounds for EOFUL and GreedyETC. The empirical validation on Llama-3-8B with TruthfulQA and HH-RLHF provides some evidence that the DDMC assumption holds in practice. However, some strong baselines are not considered. For instance, ARGS (Alignment as Reward-Guided Search, Khanov et al., 2024), as a baseline, is missing. Comparing against ARGS, especially in terms of efficiency, performance, and scalability, could help clarify why a bandit framework is preferable to a search-based alignment method like ARGS.

**Essential References Not Discussed:**

Khanov et al. "ARGS: Alignment as Reward-Guided Search" and follow-up works.

**Experimental Designs Or Analyses:**

The experimental design provides some validation of the Tokenized Bandit framework for factual correctness and preference alignment. However, the evaluation has significant gaps. While the regret bounds are well-structured, there is no explicit comparison to search-based alignment methods (e.g., ARGS) in terms of convergence speed or sample complexity. Additionally, the approach is not tested on more diverse datasets that could include safety-critical, adversarial, or instruction-following tasks to assess broader generalization. While the bandit framework is promising, these missing elements make it difficult to fully validate the empirical claims. Expanding the benchmarks and including stronger baselines would strengthen the study’s conclusions.

**Methods And Evaluation Criteria:**

The Tokenized Bandit framework (TLB & TMAB) is well-motivated for LLM decoding and alignment, as sequential token selection naturally fits a multi-armed bandit or linear bandit formulation. The authors provide a strong theoretical foundation and validate their approach on Llama-3-8B using TruthfulQA and HH-RLHF, which are reasonable benchmarks for alignment and factual correctness. However, the evaluation has some gaps: (1) No comparison to strong decoding-time alignment baselines like ARGS (Khanov et al., 2024), making it unclear if bandit-based alignment is superior to search-based alignment; (2) Limited diversity in test datasets, as evaluations focus only on two benchmarks, without testing broader alignment scenarios (e.g., safety-critical or adversarial tasks). While the evaluation is a good start, expanding it to stronger baselines and additional datasets would strengthen confidence in the method’s effectiveness.

**Other Comments Or Suggestions:**

Overall, the paper is well-written. The problem is also well-defined and the proposed solution is clear.

**Other Strengths And Weaknesses:**

Please refer to the above sections.

**Questions For Authors:**

Some of the questions that came to mind as I read the paper are detailed below. I am asking these questions in order to clarify/justify theoretical generalizability, empirical competitiveness, computational efficiency, and real-world robustness of Tokenized Bandits for LLM decoding and alignment.

1. How does the regret bound of EOFUL and GreedyETC compare to search-based decoding strategies like ARGS (Khanov et al., 2024) and variants?
2. Why is there no comparison to ARGS or reinforcement learning-based decoding strategies (e.g., PPO in RLHF)?
3. What is the computational efficiency of Tokenized Bandits compared to PPO-based RLHF and ARGS?
4. A well-known disadvantage for LinUCB is its computational complexity since it needs to incorporate historical information when initializing observation matrices and conducting matrix multiplication. Will the proposed approach inherits similar shortcomings?

**Relation To Broader Scientific Literature:**

This paper builds on foundational work in multi-armed and linear bandits, adapting these frameworks to LLM decoding and alignment. It builds upon prior work on bandits to introduce Tokenized Bandit models (TLB and TMAB) that incorporate token-wise sequential decision-making. The Diminishing Distance with More Commons (DDMC) assumption is conceptually similar to structural constraints in regret minimization (Abbasi-Yadkori et al., 2011) but is newly applied to LLM decoding. Additionally, the paper is highly relevant to decoding-time alignment approaches such as ARGS (Khanov et al., 2024), which formulates alignment as a reward-guided search problem rather than reinforcement learning. However, the authors do not compare their work directly to search-based or reinforcement learning-based decoding strategies (e.g., PPO in RLHF), leaving a gap in understanding how bandit-based decoding fares against other methods.

**Theoretical Claims:**

The paper provides rigorous mathematical proofs supporting its core claims, particularly: (1) Learning is impossible without structural assumptions in tokenized bandit settings, (2) Greedy decoding can be optimal under the Diminishing Distance with More Commons (DDMC) assumption, and (3) EOFUL (for TLB) and GreedyETC (for TMAB) achieve sublinear regret bounds. The derivations are logically consistent, following standard bandit theory techniques, and the regret bounds align with known results in linear and multi-armed bandits. However, the DDMC assumption lacks a formal justification for why it holds across different LLMs and datasets, relying primarily on empirical verification. To strengthen the theoretical foundation, the paper should provide (1) a more formal justification of DDMC beyond empirical trends.

---

> ### Author Rebuttal · Authors · 2025-03-30
>
> We truly appreciate your insightful comments.
> We will first make a general remark on our main contribution, and then answer each comment.
>
> # General remark on main contribution
>
> First, the main focus of our paper is to provide a **theoretical foundation** of tokenized versions of multi-armed bandits / linear contextual bandits, inspired by the tokenized decoding nature of the LLMs and applications such as LLM alignment / decoding.
> As shown by our impossibility results (Prop 3.3 and Prop 4.1), we reveal that some assumptions are necessary for theoretical guarantees and we provide a fairly reasonable assumption along with a relaxed version of it (Appendix B) for that sake.
>
> The need for certain assumptions in our setting (to learn a sequence function) are further justified by the fact that the seminal and  related paper by Coquelin and Munos 2007 (see our Appendix C for more comparisons) impose a **rather stronger assumption** in the tree structure to obtain efficient algorithms.
> They study bandit-based methods for tree search, motivated by real-world applications such as Go, and the objective is to learn a sequence function to maximize cumulative rewards.
> They show that without any structural assumption on the tree, exponential dependency on the depth of the tree in regret is inevitable.
> They pose a rather strong ‘smoothness’ assumption to obtain efficient algorithms.
> Such assumption was deemed innocuous in their setting, though it was not even empirically validated and not intuitively justified.
> Likewise, we impose an intuitive assumption tailored to our setting, validate it empirically, and provide a relaxed version.
> We also conduct more experiments as per the reviewer’s request, which will be explained below.
>
> # Comparison to ARGS / PPO in RLHF
>
> Thanks for the pointer!
> Regarding considering ARGS as a benchmark, the framework of ARGS cannot directly be applied in our setting, since ARGS assumes it have **access to an external reward function** and decode based on an weighted average between the original LLM’s probability and the reward.
> Our framework, however, learns the latent reward function by observing user feedback via repeated interaction, without assuming such access.
>
> Note that our regret is computed with respect to the offline benchmark that knows the latent parameter $\theta$ in hindsight, i.e. which can access the external reward function directly.
> Thus, in some sense, ARGS can be thought of as the **offline algorithm** that knows the latent parameter in hindsight.
>
> Regarding the reviewer’s comment in comparison to PPO in RLHF, we remark that our application on LLM alignment considers a **frozen LLM** that could not be retrained (e.g., for proprietary models not allowed to train or the user does not have the budget to train an LLM)" and tries to align the LLM in decoding time while learning the latent function that represents the portion of misalignment, so we think a direct comparison to PPO is irrelevant.
>
> We also refer to **our response to Reviewer 9ZG7** for numerical evaluation of our algorithm.
>
> # More justification on DDMC
>
> Our DDMC assumption, or at least a relaxed version of it, is quite reasonable intuitively.
> For instance, in extreme cases, if we append a lot of common tokens in each of two sequences of the same length, one could expect that the user will realize a similar experience of reading them.
> Further, as we discussed in L#234, page 5, this shares similar intuition from the widely adopted submodularity on set functions.
>
> Also, we conducted **more experiments** on more datasets and utility functions.
> It can be found in https://anonymous.4open.science/r/temp-03AC.
> In new experiments, we first tested DDMC assumption on two more datasets: AdvBench (standard jailbreak benchmark) and just-eval-instruct that contains many prompts of various tasks.
> Further, we verified DDMC assumptions beyond our linear utility function, with L1/3/4 distance.
> Overall, we observe a tendency that appending more common tokens decreases the utility gaps.
> We will certainly add these in the revision.
>
> # More questions
>
> ## 1. How does the regret bound of EOFUL..
>
> As noted, frameworks like ARGS assume access to the reward function, so we cannot compare regrets.
>
> ## 2. Why is there no comparison…
>
> We hope responses above clarified your question.
>
> ## 3. What is the computational efficiency...
>
> Since PPO-based RLHF retrains the LLM itself, our approach has significantly less computational burden. Further, it is unfair to compare the computational complexity of our works to ARGS, as ARGS assumes access to the external reward function whereas our approaches need to learn it.
>
> ## 4. A well-known disadvantage of…
>
> Good point! Similar to LinUCB, one can apply the standard lazy update to reduce the computational burden so that it needs to recompute the estimator only $O(log T)$ times. We refer to Section 5.1 in Abbasi-Yadkori et al., 2011 for more details. We will add a brief discussion on it in our revision.

---

> > ### Comment · Reviewer_3dxc · 2025-04-06
> >
> > I thank the authors for their rebuttal. Although I would have loved to see more detailed responses, I am happy overall with the rebuttal. I am maintaining my score.

---

### Decision · Program_Chairs · 2025-05-01

**Decision:**

Accept (poster)

**Comment:**

Three of four reviews vote to “weak accept” this paper while the fourth review is a weak reject. I looked at all the four reviews; some of the reviews raise great points in support of the paper. For instance, the reviews recognize the value of the paper in its motivation and the solid theoretical contributions presented. The experimental analysis provides useful validation, although reviews also note significant gaps and areas for improvement.

Reviewer djHj (who votes “weak reject”) raised three main concerns: (1) strong sequential dependencies of LLM generation (2) Individual reward of token and (3) comparison with other benchmarks. The authors respond to these concerns in rebuttal and answer most of the questions in two round of discussion. After initial engagement, the reviewer did not confirm if they felt satisfied, dissatisfied with all of the rebuttal provided.